

**Inter-comparison Review of IPWV retrieved from INSAT-3DR Sounder, GNSS & CAMS Reanalysis Data**

Ramashray Yadav, Ram Kumar Giri and Virendra Singh

Satellite Meteorology Division, India Meteorological Department, Ministry of Earth Sciences

New Delhi-110003

**Abstract:**

The spatiotemporal variations of integrated precipitable water vapor (IPWV) are very important to understand the regional variability of water vapour. Traditional in-situ measurements of IPWV in Indian region are limited and therefore the performance of satellite and Copernicus Atmosphere Meteorological Service (CAMS) retrieval with Indian Global Navigation Satellite System (GNSS) taking as reference has been analyzed. In this study the CAMS reanalysis data one year (2018), Indian GNSS and INSAT-3DR sounder retrievals data for one & half years (January-2017 to June-2018) has been utilized and computed statistics. It is noticed that seasonal correlation coefficient (CC) values between INSAT-3DR and Indian GNSS data mainly lie within the range of 0.50 to 0.98 for all the selected 19 stations except Thiruvanathpuram (0.1), Kanyakumari (0.31), Karaikal (0.15) during monsoon and Panjim (0.2) during post monsoon season respectively. The seasonal CC values between CAMS and INSAT-3DR IPWV are ranges 0.73 to .99 except Jaipur (0.16) & Bhubneshwar (0.29) during pre-monsoon season, Panjim (0.38) during monsoon, Nagpur (0.50) during post-monsoon and Dibrugarh (0.49) Jaipur (0.58) & Bhubaneswar (0.16) during winter season respectively .The root mean square error (RMSE) values are higher under the wet conditions (Pre Monsoon & Monsoon season) than under dry conditions (Post Monsoon & Winter season) and found differences in magnitude and sign of bias of INSAT-3DR, CAMS with respect to GNSS IPWV from station to station and season to season.

This study will help to improve understanding and utilization of CAMS and INSAT-3DR data more effectively along with GNSS data over land, coastal and desert locations in terms of seasonal flow of IPWV which is an essential integrated variable in forecasting applications.

**Keywords**: Indian Satellite -3DR (INSAT-3DR), Integrated Precipitable Water Vapour (IPWV), Copernicus Atmospheric Monitoring Service (CAMS) & Global Navigation Satellite System (GNSS).

## Introduction

Integrated precipitable water vapor (IPWV) is a meteorological factor that shows the amount of water vapour contained in the column of air per unit area of the atmosphere in terms of the depth of liquid (Viswanadham et al., 1981). This parameter has great importance in all studies related to the atmosphere and its properties throughout the year and in all seasons. The assessment of IPWV is done in many ways as in situ, model based or through remote sensing measurements. The in situ measurements have limited coverage, expensive and require maintenance of all the time. Remote sensing instruments, especially absorption in the infrared and microwave region of the solar spectrum have wide coverage, cheaper, almost maintenance free but needs to validate their retrieval performance and inter comparison before applying in the operational meteorological service domain. Similarly, model based data have limitations to capture the localized features of convection due to sparseness or very few numbers of the quality controlled observational data over that region. Water vapour content present in the atmosphere, one of the most influential constituents of the atmosphere, is responsible for determining the amount of precipitation that a region can receive (Trenberth et al., 2003). The absorptions of surface radiation depends on wavelength and water vapor content. Each absorbing water vapour molecule emits radiation according to Planck's law, mainly depending on its temperature and the extent of absorption differs depending on the wavelength, the satellite sees different levels of atmosphere.

In Global Ozone Monitoring Experiment (GOME) and Scanning Imaging Absorption Spectrometer for Atmospheric CHartography (SCIAMACHY), both used the principle of differential optical absorption spectroscopy in red spectral range of IPWV retrieval (Beirle et al, 2018). Atmospheric Infrared sounder is a hyper spectral instrument which collects radiances in 2378 IR channels with wavelength ranging from 3.7 to 15.4 µm. Cloud cleared radiances of AIRS were utilized in the retrieval of column integrated water vapour which is contributed by a number of channels having different sensitivity towards water vapour content present in the atmosphere. (Aumann et al., 2003). Moderate Resolution Imaging Spectroradiometer (MODIS) utilized infrared algorithm employs ratios of water vapor absorbing channels at 0.905 µm, 0.936 µm, and 0.940 µm with atmospheric window channels at 0.865 µm and 1.24 µm estimated the precipitable water vapour (Kaufman and Gao, 1992).

The uncertainties in the retrieval of precipitable water vapor from satellites (like errors of calibration of channels, viewing geometry, radiative transfer in the forward models) are already addressed by previous studies (Ichoku et al., 2005 for MODIS, Noel et al., 2008 for GOME-2 and SCIAMACHY, Susskind et al., 2003, 2006 for AIRS). Wagner et al., 2006 studied GOME data for the period of 1996-2002 and reported globally and yearly averaged 2.8 ±0.8% increase of total column precipitable water (excluding the ENSO period).

The retrievals from reanalysis data sets Modern-Era Retrospective analysis for Research and applications-2 (MERRA-2) Gelaro et al., 2017 , Climate Forecast System Reanalysis (CFSR) (Saha et al., 2010) Data Archive at https://rda.ucar.edu/pub/cfsr.html utilized 3d-var data

assimilation techniques and well captured the interannual variations of precipitable water vapour
in the south of the Central Asia (Jiang et al., 2019). The study carried out by Berrisford et al., 2011,
found ERA interim data set is superior in quality than ERA 40 during the period 1989-2008.
Ramashray et al., 2020 carried out the validation of Indian GNSS IPWV with GPS Sonde data for
the period of June 2017 to May 2018 over Indian region and found reasonably well in agreement
with in situ observations. In situ Radiosonde observations generally suffer spatiotemporal
inhomogeneity errors and differences in relative humidity measured by different sensors. In this
study he brought out positive bias less than 4.0 mm for 7 stations, correlation coefficient greater
than 0.85 and RMSE less than 5.0 mm for all 09 collocated GPS sonde stations. In this direction
the work carried out by Turner et al., 2003, 5 % dry bias with Microwave Radiometer and Vaisala
RS80-H will be very useful while dealing with such Radiosonde observations. Miloshevich et al.,
2009, found a similar limitation of Relative Humidity measurement with Vaisala RS92 Radio
sonde and derived an empirical correction to remove the mean bias error, yielding bias uncertainty
is independent of height.
The study carried out by Falaiye et al., 2018 is very important for considering the conventional
data from long term observing stations of Indian domain along with the available model to
establish the similar empirical relationship of getting the precipitable water vapour. This will also
support to generate improved climatological mean especially over the remote regions.
Geo satellites have higher temporal resolution and continuous coverage and are important for
monitoring the extreme weather events. Polar satellites have higher advantage higher spatial
resolution and can operate both cloudy and non-cloudy conditions more effectively as compared
to Geo satellites. Courcoux and Schroder et al., 2013, worked out the accuracies of Satellite
Application Facility on Climate Monitoring (CMSAF) satellite Advanced Television and Infrared
Observation Satellite Operational Vertical Sounder (ATOVS) precipitable water vapour of about
2-4 mm with respect to radiosonde and Atmospheric Infrared Sounder (AIRS) data both over land
and ocean with resolution of 0.5º x 0.5º.
Geo-stationary Earth Orbit (GEO) satellites can produce data more timely and frequently. The
retrieved high temporal resolution, Integrated Precipitable Water vapour (IPWV) from GEO
satellites sensor data can be utilized to monitor pre-convective environments and predict heavy
rainfall, convective storms, and clouds that may cause serious damage to human life and
infrastructure (Martinez et al., 2007; Liu et al., 2019; Lee et al., 2015). At present two advanced
Indian geostationary meteorological satellites INSAT-3D (launched on 26 July, 2013) and INSAT-
3DR on 6 September, 2016) with similar sensor characteristics are orbiting over Indian Ocean
region and are placed at 82° E and 74° E respectively. INSAT -3D & INSAT-3DR both satellites
are equipped with the infrared sounders with 19 channels, which are used to provide
meteorological parameters like the profiles of temperature, humidity and ozone, atmospheric
stability indices, atmospheric water vapor, etc. at 1 hour (sector A)  and 1.5 hour (sector B)
intervals (Kishtawal et al., 2019).   Temperature and humidity (T-q profile) is used to retrieve
thermodynamic indices which is useful in analyzing the strength and severity of severe weather
events. Therefore, IPWV is one of the critical variables used by forecasters when severe weather
conditions are expected (Lee et al., 2016). Copernicus Atmosphere Monitoring Service (CAMS)
global reanalysis (EAC4) latest data set of atmospheric composition has been built at approximate
80 km resolution with improved biases and consistent with time. (Inness et al., 2019).The concept
of GNSS meteorology was first introduced by Bevis et al.,1992, 1994 and Businger et al., 1992
and IPWV data were estimated from Global Navigation Satellite System (GNSS) observations. In
this study we have taken 19 Indian GNSS stations (10 inland, 8 coastal and 1 desert) or sites for
study. Earlier studies (Jade et al., 2005; Jade and Vijayan et al., 2008; Puviarasan et al., 2014) of
water vapour over the Indian subcontinent and surrounding ocean have shown strong seasonal
variations.
The behavior of coastal regions are generally different from inland and  desert stations as coastal
regions  greatly influenced moisture advection from breezing of the seas, which is the cause of the
continuous increment of IPWV even after the air temperature decreased (Ortiz de Galisteo et al.,
123  2011).

Perez-Ramirez et al., 2014, compared Aerosol Robotic Network (AERONET) precipitable water
vapour retrievals from Sun photometers with radiosonde, ground based Microwave radiometry,
GPS and found a consistent dry bias approximately 5-6 % with total uncertainties of 12-15 % in
the retrievals of precipitable water vapour from AERONET. The study Perez-Ramirez et al.,
(2019) clearly brought out the importance of Maritime Aerosol. Network (MAN) in retrieving the
precipitable water vapour over remote oceanic areas. The reanalysis model estimates have very
good agreement with MAN with mean differences of ~ 5 % and standard deviation of ~15 % under
clear sky conditions. The work done in the past by Smirnov et al., 2004, 2011,  in retrieving the
precipitable water vapour from aerosol network data especially for marine areas is very helpful to
carry out further studies in future with INSAT-3DR satellite observations over oceanic areas.
The present study have two fold objectives (1) Inter-comparison of CAMS and INSAT-3DR, IPW
retrievals with Indian GNSS stations by taking GNSS reference and (II) performance in  the
retrievals CAMS and INSAT-3DR sounder for both land and ocean regions. This analysis will be
very useful to know about the satellite and reanalysis uncertainties and their improvements over
place to place and season to season. It will also further improve and help the forecasters to use
models as well as INSAT-3DR data sets with confidence as these are available over wide spatial
coverage as compared to low density of GNSS network data over Indian domains.
**2. Methodology and Data collection**
The measured Integrated Precipitable Water Vapour (IPWV) from the India Meteorological
Department (IMD) GNSS network with 15 minute temporal resolution data are used for the
comparison of INSAT-3DR geostationary satellite IPWV products and CAMS reanalysis IPWV
data. The INSAT-3DR data scans are each of one hour intervals from January-2017 to June-2018.
These measured and derived IPWV products are arranged as co-location of both temporal and
spatial. The spatial views of the observational locations of GNSS and along with INSAT-3DR
IPWV annual mean values are shown in Figure 2. The number of observational points (N) of each
GNSS, INSAT-3DR and CAMS reanalysis of each station with its latitude, longitude are shown
in Table 2. Here, winter season is considered as December, January and February months; pre
monsoon season is considered as March, April and May; monsoon season in June, July and August
months; finally post monsoon season is considered as September, October and November  months.
**2.1 IMD IPWV observation network**
The ground based GNSS IPWV estimated at a high temporal sampling (15 minute) data (January
2017- June 2018) of Indian GNSS network is processed at satellite division of India
Meteorological Department, Lodi Road, New Delhi. The data is processed daily by using the
Trimble Pivot Platform (TPP) software.
The data is used operationally and archive as daily, weekly, monthly as well as seasonal basis for
future utilization and dissemination to the users, researchers as per the official norms. If we reduce
the cut off angle from 5° multipath effect will occur and introduce inaccuracy in the IPWV
estimation. An elevation angle of greater than 5° is set for all stations to avoid the satellite geometry
change and multipath effects. This is an optimum setting as a higher cut off angle ($> 5°$) may
introduce dry bias in the IPWV estimation and notable 0.8 mm error in IPWV (Emardson et al.,
1998). The other possible sources of error associated with GNSS data are mean temperature of
atmosphere, dynamical pressure and isotropic errors. These errors will vary with location and time
of observations.
**2.2 Integrated Precipitable Water Vapour retrievals from INSAT-3DR Sounder data**
The Sounder payload of the INSAT-3DR satellite has the capability to provide vertical profiles of
temperature (40 levels from surface to ~ 70 km) and humidity (21 levels from surface to ~ 15 km)
from surface to top of the atmosphere. The Sounder has eighteen narrow spectral channels in
shortwave infrared, middle infrared and long wave infrared regions and one channel in the visible
region. The ground resolution at nadir is $10 \times 10$ km for all nineteen channels. Specifications of
sounder channels are given in Table 1. Vertical profiles of temperature and moisture can be derived
from radiances in these 18 IR channels, using the first guess from numerical weather prediction
(NWP) model data. INSAT-3DR sounder channels brightness temperature values are averaged
over a number of fields of view (FOVs) prior to application of retrieval algorithm. Based on this,
average vertical profiles are retrieved at 30 x 30 km ($3 \times 3$ pixels) for each cloud free pixel.
As INSAT-3DR IPWV is sensitive to the presence of clouds in the field of view (limitation of
Infra-red sounder sensors), hence the IPWV values collected under clear sky conditions were used
in this study. Atmospheric profile retrieval algorithm for INSAT-3DR Sounder is a two-step
approach. The first step includes generation of accurate hybrid first guess profiles using a
combination of statistical regression retrieved profiles and model forecast profiles. The second
step is nonlinear physical retrieval to improve the resulting first guess profile using Newtonian
iterative method. The retrievals are performed using clear sky radiances measured by sounder
within a 3x3 field of view (approximately 30x30 km resolution) over land for both day and night
(similar to INSAT-3D ATBD, 2015). Four sets of regression coefficients are generated, two sets
for land and ocean daytime conditions and the other two sets for land and ocean night-time
conditions using a training dataset comprising historical radiosonde observations representing
atmospheric conditions over INSAT-3DR observation region. Integrated Precipitable Water
Vapour in mm can be given as:
$$PWV = \int_{p_1}^{p_2} \frac{q}{g\rho_w} \, dp$$

Where, 'g' is the acceleration of gravity, $p_1$ = surface pressure, $p_2$ = top of atmosphere pressure
(i.e. about 100 hPa beyond which water vapour amount is assumed to be negligible). Unit of
precipitable water is mm depth of equal amount of liquid water above a surface of one square
meter. IMD is computing IPWV from 19 channel sounder of INSAT-3DR in three layers i.e. 1000-
900 hPa, 900-700 hPa, 700-300 hPa and total PWV in the vertical column of atmosphere stretching
from surface to about 100 hPa during cloud free condition. Monsoon, severe weather, cloudy
condition puts the limitation for sounder profile (Venkat Ratnam et al., 2016). The GNSS and
INSAT-3DR retrieved IPWV values are matched at every hour.

## 2.3 Scan Strategy of INSAT-3DR Sounder

The Sounder measures radiance in eighteen infrared (IR) and one visible channel simultaneously
over an area of area of 10 km x 10 km at nadir every 100 ms. Using a two-axes gimballed scan
mirror, this footprint can be positioned anywhere in the field of regard (FOR)- 24º (E-W) x 19º
(N-S).  To Sound the entire globe area of 6400 km x 6400 km in size, it takes almost three hours.
A scan program mode allows sequential sounding of a selected area with periodic space and
calibration looks. In this mode, a 'frame' consisting of multiple 'blocks' of the size 640 km x 640
km, can be sounded. The selected frame can be placed anywhere within a 24º (E-W) x 19º (N-S)
(similar to INSAT-3D ATBD, 2015).  An optimized scan strategy of sounder payload is worked
out involving all stakeholders in such a way Indian land region sector-A data covered up on hourly
basis and Indian Ocean region Sector-B data covered up on one & half hourly basis as shown in
Figure 1. The full aperture internal Black-body calibration is performed every 30 min or on
command based whenever required. The sounder payload has a provision to be carried out on
board IR calibration, in which the scan mirror pointed towards the space look to measure the
radiances then pointed to the internal blackbody present on the payload for measuring its radiances.
There is also a provision to measure the temperature of the internal black body. All these data sets
are transmitted along with video data of payload.  During the processing at ground, the data
collected during on board calibration are used to generate the calibration look up table for each
scan. This enables the derivation of vertical profiles of temperature and humidity more accurately.
These vertical profiles can then be used to derive various atmospheric stability indices and other
parameters such as atmospheric water vapor content and total column ozone amount. The products
derived over sector-A data are used for weather forecasting on operational basis and products
derived over sector-B are used for assimilation in NWP model.

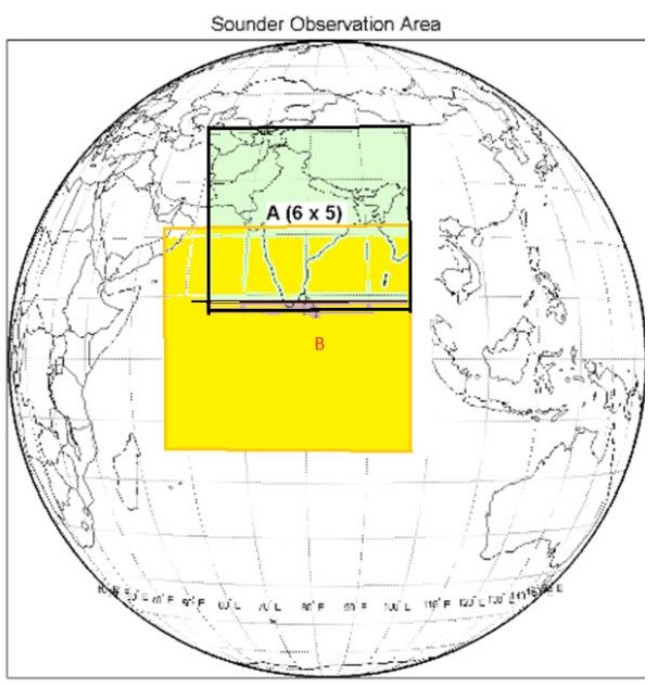




Sector-A                                                                          Sector-B
0300, 0400, 0500 UTC-INSAT-3DR                                    0000, 0130 UTC INSAT-3DR
Figure 1.Scan Strategy and Area of Coverage of INSAT-3DR Sounder payload.

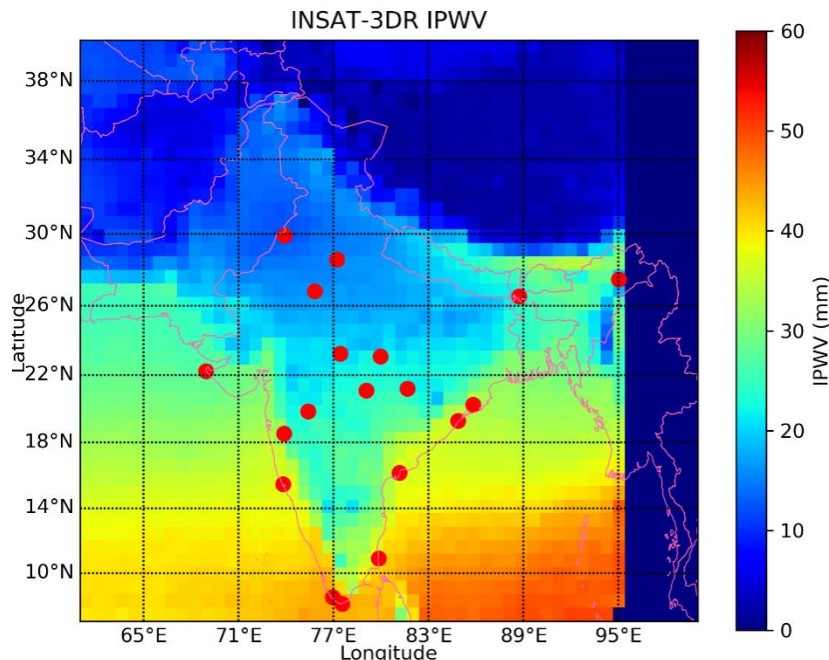


Figure 2. The annual mean of IPWV over India retrieved from INSAT- 3DR during the year of
2018.The geographical distribution of 19 GNSS stations (filled Red color circles).
Table 1. INSAT-3DR Sounder channel specifications

| INSAT-3DR Sounder Channels Characteristics | | | | |
|---|---|---|---|---|
| Detector | Channel No. | Central Wavelength (μm) | Principal absorbing gas | Purpose |
| Long wave | 1 | 14.67 | $CO_2$ | Stratosphere temperature |
| | 2 | 14.32 | $CO_2$ | Tropopause temperature |
| | 3 | 14.04 | $CO_2$ | Upper-level temperature |
| | 4 | 13.64 | $CO_2$ | Mid-level temperature |
| | 5 | 13.32 | $CO_2$ | Low-level temperature |
| | 6 | 12.62 | water vapor | Total precipitable water |
| | 7 | 11.99 | water vapor | Surface temp., moisture |
| Mid wave | 8 | 11.04 | Window | Surface temperature |
| | 9 | 9.72 | Ozone | Total ozone |
| | 10 | 7.44 | water vapor | Low-level moisture |

| | 11 | 7.03 | water vapor | Mid-level moisture |
|---|---|---|---|---|
| | 12 | 6.53 | water vapor | Upper-level moisture |
| Short wave | 13 | 4.58 | $N_2O$ | Low-level temperature |
| | 14 | 4.53 | $N_2O$ | Mid-level temperature |
| | 15 | 4.46 | $CO_2$ | Upper-level temperature |
| | 16 | 4.13 | $CO_2$ | Boundary-level temp. |
| | 17 | 3.98 | Window | Surface temperature |
| | 18 | 3.76 | Window | Surface temp., moisture |
| Visible | 19 | 0.695 | Visible | Cloud |


Table 2. List of GNSS stations (latitude, longitude, height) and location environment

| S.No | Station | Station code | Long | Lat | Ellipsoid Height(m) | Environment |
|---|---|---|---|---|---|---|
| 1 | Aurangbad | ARGD | 75.39 | 19.87 | 528.13 | Inland |
| 2 | Bhopal | BHPL | 77.42 | 23.24 | 476.22 | Inland |
| 3 | Dibrugarh | DBGH | 95.02 | 27.48 | 55.76 | Inland |
| 4 | Delhi | DELH | 77.22 | 28.59 | 165.06 | Inland |
| 5 | Jabalpur | JBPR | 79.98 | 23.09 | 355.09 | Inland |
| 6 | Jaipur | JIPR | 75.81 | 26.82 | 335.37 | Inland |
| 7 | Jalpaiguri | JPGI | 88.71 | 26.54 | 37.41 | Inland |
| 8 | Pune | PUNE | 73.88 | 18.53 | 487.72 | Inland |
| 9 | Raipur | RIPR | 81.66 | 21.21 | 245.56 | Inland |
| 10 | Nagpur | NGPR | 79.06 | 21.09 | 253.57 | Inland |
| 11 | Dwarka | DWRK | 68.95 | 22.24 | -40.12 | Coastal |
| 12 | Gopalpur | GOPR | 84.87 | 19.3 | -15.94 | Coastal |
| 13 | Karaikal | KRKL | 79.84 | 10.91 | -79.07 | Coastal |
| 14 | Kanyakumari | KYKM | 77.54 | 8.08 | -49.23 | Coastal |
| 15 | Machilipattnam | MPTM | 81.15 | 16.18 | -61.07 | Coastal |
| 16 | Panjim | PNJM | 73.82 | 15.49 | -23.04 | Coastal |
| 17 | Thiruvanathpuram | TRVM | 76.95 | 8.5 | -18.44 | Coastal |
| 18 | Bhubneshwar | BWNR | 85.82 | 20.25 | -16.72 | Coastal |
| 19 | Sriganganagar | SGGN | 73.89 | 29.92 | 132.17 | Desert |




## 2.4 Copernicus Atmosphere Monitoring Service (CAMS) reanalysis data

The CAMS reanalysis was produced using 4DVar data assimilation in European Centre for Medium Range Weather Forecasts (ECMWF) Integrated Forecasting System (IFS), with 60 hybrid sigma / pressure (model) levels in the vertical, with the top level at 0.1 hPa (https://ads.atmosphere.copernicus.eu/cdsapp#!/search?type=dataset). Atmospheric data are available on these levels and they are also interpolated to 25 pressure levels, 10 potential temperature levels and 1 potential vorticity level (Inness et al., 2019).This new reanalysis data set has horizontal resolution of about 80 km (0.75° x 0.75°), smaller biases for reactive gases and aerosols, improved and more consistent with time as compared to earlier versions. INSAT-3DR Data set has horizontal resolution at 30 x 30 km (3 × 3 pixels) for each cloud free pixel. Collocation match up has been created at 0.75° x 0.75° (about 80 km) spatial resolution for comparison and performance of INSAT-3DR data with CAMS reanalysis data using bilinear interpolation technique. Temporal domains are selected at 00, 03, 06, 09, 12, 15, 18, 21 UTC time interval for Indian GNSS along with INSAT-3DR at 03, 09, 15, 21 UTC for performance analysis. The CAMS reanalysis IPWV retrievals are interpolated to different geographical locations of 19 GNSS observations. We have used nearest neighbor interpolation techniques to interpolate CAMS reanalysis with GNSS data. In this method we evaluate each station to determine the number of neighboring grid cells in 0.75° x 0.75° box that surround the GNSS station and contain at least one valid CAMS reanalysis data. CAMS data is capable of capturing large scale features of moisture flow which help the forecasters in predicting large scale weather systems such as western disturbances, cyclonic storms, monitoring of monsoon and other associated weather events affecting throughout the year in Indian domain.

## 2.5 Analysis of statistical skill scores

The collocated comparison statistics with matchup data set is used to evaluate the statistical performance of retrievals of INSAT-3DR and CAMS with respect to GNSS IPWV over Indian region.

The statistical metrics used for quantitative evaluation are, linear correlation coefficient (CC), Standard Deviation (SD), Bias and Root Mean Square Error (RMSE).The computation of above said statistical metrics are given below:

Let, $O_i$ represents the $i^{th}$ observed value of INSAT-3DR or CAMS reanalysis data and $M_i$ represents the $i^{th}$ GNSS IPWV value for a total of n observations.

Mean bias (MB)

$$MB = \frac{1}{n}\sum_{i=1}^{N}(O_i - M_i)$$


Root Mean Squared Error (RMSE)

$$RMSE = \sqrt{\frac{1}{N}\sum_{i=1}^{N}(O_i - M_i)^2}$$

Correlation Coefficient (CC)

$$CC = \frac{N(\sum_{i=1}^{N} M_i O_i) - (\sum_{i=1}^{N} M_i)(\sum_{i=1}^{N} O_i)}{\sqrt{[N\sum_{i=1}^{N} M^2_i - (\sum_{i=1}^{N} M_i)^2][N\sum_{i=1}^{N} O^2_i - (\sum_{i=1}^{N} O_i)^2]}}$$


Standard Deviation (SD)

$$SD = \sqrt{\left\{\frac{[N\sum_{i=1}^{N}(M_i - \overline{M})^2][N\sum_{i=1}^{N}(O_i - \overline{O})^2]}{N}\right\}}$$


## 2.6 INSAT-3DR and GNSS retrievals matchup criteria

The assessment of accuracy of INSAT-3DR satellite retrieved IPWV with 19 GNSS stations in
different geographical locations which are located in coastal, inland and desert regions over the
Indian subcontinent and are shown in Table 2. The GNSS IPWV data sampled every 15 minute
and to maintain consistency with INSAT-3DR retrievals that are available every one hour interval
of time over the Indian region for the period 1st January 2017 to 30th June 2018 have been utilized.
Matchup data sets for were prepared for INSAT-3DR and GNSS IPWV as per the following
criteria
(1) To reduce the local horizontal gradient arising in IPWV, The absolute distance between the
position of the GNSS stations locations are set within the 0.25° latitude and longitude of the
INSAT-3DR retrievals in the region surrounding the stations.
(2) The temporal resolution selected of INSAT-3DR and 19 GNSS observations is within 30 min
time interval depending on retrievals and the location of the GNSS stations.
(3) The INSAT-3DR IPWV retrievals are interpolated to different geographical locations of 19
GNSS observations.



Table 3. Statistical analysis of IPWV retrievals from INSAT-3DR & GNSS data (January-2017
& June-2018).


| S. No | Station | N | MB (mm) | RMSE (mm) | R |
|---|---|---|---|---|---|
| 1 | ARGD | 2318 | -0.99 | 4.83 | 0.85 |
| 2 | BHPL | 791 | 3.48 | 5.88 | 0.93 |
| 3 | DBGH | 688 | -3.02 | 12.38 | 0.72 |
| 4 | DELH | 1880 | -1.58 | 4.53 | 0.89 |
| 5 | NGPR | 2032 | -0.10 | 4.32 | 0.89 |
| 6 | JBPR | 952 | 1.96 | 4.39 | 0.93 |
| 7 | JIPR | 1576 | 0.46 | 4.26 | 0.88 |
| 8 | JPGI | 1551 | 2.25 | 8.10 | 0.75 |
| 9 | PUNE | 567 | 0.69 | 6.18 | 0.83 |
| 10 | RIPR | 1849 | 0.71 | 4.01 | 0.84 |
| 11 | BWNR | 1443 | 1.51 | 5.61 | 0.88 |
| 12 | DWRK | 2628 | 2.93 | 7.10 | 0.85 |
| 13 | GOPR | 1850 | 0.76 | 7.59 | 0.82 |
| 14 | KRKL | 1128 | 0.52 | 6.59 | 0.88 |
| 15 | KYKM | 1574 | 1.91 | 7.21 | 0.80 |
| 16 | MPTM | 1747 | 3.12 | 7.29 | 0.81 |
| 17 | TRVM | 905 | 0.01 | 7.56 | 0.76 |
| 18 | PNJM | 1396 | -2.93 | 9.28 | 0.67 |
| 19 | SGGN | 1040 | -1.41 | 4.42 | 0.88 |


Table 4. Statistical seasonal analysis of retrievals of IPWV from INSAT-3DR and GNSS data

| Station | Season | N | MB (mm) | RMSE (mm) | R |
|---|---|---|---|---|---|
| ARGD | Pre Monsoon (MAM) | 1129 | -2.10 | 4.14 | 0.86 |
| | Monsoon (JJA) | 73 | -0.53 | 5.50 | 0.49 |
| | Post Monsoon (SON) | 271 | 3.02 | 6.23 | 0.90 |
| | Winter (DJF) | 845 | -0.84 | 5.10 | 0.67 |
| BHPL | Pre Monsoon (MAM) | 69 | -0.49 | 3.81 | 0.77 |

| | | | | | |
|---|---|---|---|---|---|
| | Monsoon (JJA) | 78 | 2.10 | 7.73 | 0.64 |
| | Post Monsoon (SON) | 339 | 5.23 | 6.96 | 0.93 |
| | Winter (DJF) | 305 | 2.78 | 4.16 | 0.95 |
| DBGH | Pre Monsoon (MAM) | 214 | -1.96 | 6.69 | 0.72 |
| | Monsoon (JJA) | 83 | -12.39 | 14.71 | 0.64 |
| | Post Monsoon (SON) | 79 | -22.52 | 27.74 | -0.28 |
| | Winter (DJF) | 312 | 3.68 | 7.39 | 0.48 |
| DELH | Pre Monsoon (MAM) | 793 | -1.44 | 3.98 | 0.85 |
| | Monsoon (JJA) | 84 | -5.79 | 7.90 | 0.92 |
| | Post Monsoon (SON) | 230 | -0.76 | 5.13 | 0.92 |
| | Winter (DJF) | 773 | -1.51 | 4.36 | 0.79 |
| NGPR | Pre Monsoon (MAM) | 772 | -1.42 | 4.06 | 0.85 |
| | Monsoon (JJA) | 25 | 0.39 | 5.41 | 0.57 |
| | Post Monsoon (SON) | 254 | 1.08 | 5.86 | 0.90 |
| | Winter (DJF) | 981 | 0.61 | 4.00 | 0.83 |
| JBPR | Pre Monsoon (MAM) | 438 | 1.51 | 4.79 | 0.84 |
| | Monsoon (JJA) | 11 | -4.05 | 4.43 | 0.92 |
| | Post Monsoon (SON) | 50 | 1.89 | 3.94 | 0.98 |
| | Winter (DJF) | 453 | 2.54 | 4.02 | 0.94 |
| JIPR | Pre Monsoon (MAM) | 505 | -0.44 | 3.86 | 0.83 |
| | Monsoon (JJA) | 70 | -3.84 | 5.89 | 0.92 |
| | Post Monsoon (SON) | 383 | 1.34 | 4.48 | 0.89 |
| | Winter (DJF) | 618 | 1.13 | 4.21 | 0.71 |
| JPGI | Pre Monsoon (MAM) | 527 | -1.59 | 6.88 | 0.79 |
| | Monsoon (JJA) | 67 | -6.69 | 9.25 | 0.75 |
| | Post Monsoon (SON) | 161 | 9.43 | 10.91 | 0.65 |
| | Winter (DJF) | 796 | 4.09 | 8.07 | 0.50 |
| PUNE | Pre Monsoon (MAM) | 333 | 0.03 | 6.65 | 0.72 |
| | Monsoon (JJA) | 63 | -3.10 | 5.09 | 0.67 |
| | Post Monsoon (SON) | 170 | 3.35 | 5.54 | 0.79 |
| | Winter (DJF) | 1 | 5.90 | 5.90 | NaN |
| RIPR | Pre Monsoon (MAM) | 864 | -0.39 | 3.94 | 0.84 |
| | Monsoon (JJA) | 0 | NaN | NaN | NaN |
| | Post Monsoon (SON) | 68 | 4.83 | 6.09 | 0.75 |
| | Winter (DJF) | 917 | 1.45 | 3.88 | 0.77 |
| KRKL | Pre Monsoon (MAM) | 739 | 0.03 | 5.29 | 0.89 |
| | Monsoon (JJA) | 105 | -0.58 | 8.54 | 0.15 |
| | Post Monsoon (SON) | 31 | -1.88 | 8.54 | 0.59 |
| | Winter (DJF) | 253 | 2.68 | 8.53 | 0.63 |
| KYKM | Pre Monsoon (MAM) | 686 | 0.31 | 5.84 | 0.79 |

| | | | | | |
|---|---|---|---|---|---|
| | Monsoon (JJA) | 110 | -1.73 | 9.53 | 0.31 |
| | Post Monsoon (SON) | 155 | 0.88 | 11.21 | 0.50 |
| | Winter (DJF) | 623 | 4.56 | 6.83 | 0.88 |
| MPTM | Pre Monsoon (MAM) | 767 | 2.17 | 5.54 | 0.81 |
| | Monsoon (JJA) | 40 | 2.47 | 5.22 | 0.77 |
| | Post Monsoon (SON) | 172 | -0.43 | 13.49 | 0.48 |
| | Winter (DJF) | 768 | 4.89 | 6.94 | 0.73 |
| GOPR | Pre Monsoon (MAM) | 837 | -1.22 | 7.11 | 0.70 |
| | Monsoon (JJA) | 29 | -2.25 | 4.23 | 0.88 |
| | Post Monsoon (SON) | 253 | 1.55 | 11.41 | 0.69 |
| | Winter (DJF) | 731 | 2.87 | 6.48 | 0.72 |
| DWRK | Pre Monsoon (MAM) | 1119 | 1.42 | 7.12 | 0.62 |
| | Monsoon (JJA) | 377 | -0.93 | 5.47 | 0.78 |
| | Post Monsoon (SON) | 362 | 6.09 | 8.37 | 0.87 |
| | Winter (DJF) | 770 | 5.54 | 7.12 | 0.82 |
| PNJM | Pre Monsoon (MAM) | 878 | -4.75 | 10.27 | 0.60 |
| | Monsoon (JJA) | 46 | -0.39 | 5.76 | 0.60 |
| | Post Monsoon (SON) | 39 | -6.10 | 18.73 | 0.20 |
| | Winter (DJF) | 433 | 0.79 | 5.35 | 0.64 |
| TRVM | Pre Monsoon (MAM) | 360 | -1.85 | 6.98 | 0.75 |
| | Monsoon (JJA) | 53 | -7.05 | 11.36 | 0.10 |
| | Post Monsoon (SON) | 113 | -0.32 | 10.56 | 0.42 |
| | Winter (DJF) | 379 | 2.87 | 6.25 | 0.82 |
| BWNR | Pre Monsoon (MAM) | 441 | 0.39 | 5.71 | 0.80 |
| | Monsoon (JJA) | 12 | -5.22 | 7.37 | 0.89 |
| | Post Monsoon (SON) | 92 | 3.56 | 8.36 | 0.79 |
| | Winter (DJF) | 898 | 1.94 | 5.16 | 0.82 |
| SGGN | Pre Monsoon (MAM) | 179 | -1.23 | 3.81 | 0.79 |
| | Monsoon (JJA) | 33 | -3.96 | 5.49 | 0.91 |
| | Post Monsoon (SON) | 432 | -3.24 | 5.52 | 0.87 |
| | Winter (DJF) | 396 | 0.72 | 2.99 | 0.91 |


Table 5. Statistical analysis of IPWV retrievals from CAMS & GNSS data (January to December
311 2018)

| S.No. | Station | N | MB (mm) | RMSE (mm) | R |
|---|---|---|---|---|---|
| 1 | ARGD | 1624 | -2.72 | 3.69 | 0.97 |
| 2 | BHPL | 0 | NaN | NaN | NaN |
| 3 | DBGH | 1002 | 2.91 | 6.7 | 0.95 |
| 4 | DELH | 2345 | -1.27 | 3.09 | 0.99 |

| 5 | NGPR | 1325 | 1.99 | 9.17 | 0.88 |
|---|------|------|------|------|------|
| 6 | RIPR | 1727 | -1.94 | 3.48 | 0.98 |
| 7 | JBPR | 1483 | -1.11 | 3.25 | 0.99 |
| 8 | PUNE | 1165 | -6.69 | 7.62 | 0.96 |
| 9 | JIPR | 1483 | 0.75 | 7.19 | 0.92 |
| 10 | JPGI | 2168 | -0.68 | 3.83 | 0.98 |
| 11 | BWNR | 1240 | 7.5 | 13.59 | 0.48 |
| 12 | KRKL | 1949 | -0.9 | 3.74 | 0.96 |
| 13 | KYKM | 2145 | 0.47 | 3.33 | 0.96 |
| 14 | MPTM | 1929 | -1.3 | 3.69 | 0.97 |
| 15 | PNJM | 750 | 2.27 | 7.25 | 0.78 |
| 16 | GOPR | 1625 | -0.41 | 3.76 | 0.98 |
| 17 | DWRK | 2094 | -0.87 | 3.12 | 0.98 |
| 18 | TRVM | 2073 | -1.91 | 4.33 | 0.93 |
| 19 | SGGN | 2274 | -1.74 | 3.37 | 0.98 |



Table 6.Statistical seasonal analysis of retrievals of IPWV from CAMS and GNSS data

| Station | Season | N | MB (mm) | RMSE(mm) | R |
|---------|--------|---|---------|----------|---|
| ARGD | Pre Monsoon (MAM) | 673 | -2.09 | 3.25 | 0.93 |
|  | Monsoon (JJA) | 97 | -3.02 | 5.32 | 0.75 |
|  | Post Monsoon (SON) | 248 | -3.42 | 4.24 | 0.97 |
|  | Winter  Winter (DJF) | 606 | -3.09 | 3.6 | 0.96 |
| BHPL | Pre Monsoon (MAM) | 0 | NaN | NaN | NaN |
|  | Monsoon (JJA) | 0 | NaN | NaN | NaN |
|  | Post Monsoon (SON) | 0 | NaN | NaN | NaN |
|  | Winter (DJF) | 0 | NaN | NaN | NaN |
| DBGH | Pre Monsoon (MAM) | 261 | 5.98 | 7.48 | 0.92 |
|  | Monsoon (JJA) | 169 | 6.6 | 7.43 | 0.84 |
|  | Post Monsoon (SON) | 396 | 1.39 | 6.37 | 0.95 |
|  | Winter (DJF) | 176 | -1.76 | 5.31 | 0.49 |
| DELH | Pre Monsoon (MAM) | 719 | -0.86 | 2.83 | 0.95 |
|  | Monsoon (JJA) | 223 | 0.2 | 4.9 | 0.92 |
|  | Post Monsoon (SON) | 721 | -2.22 | 3.57 | 0.99 |
|  | Winter (DJF) | 682 | -1.19 | 1.74 | 0.97 |
| NGPR | Pre Monsoon (MAM) | 192 | -0.53 | 2.27 | 0.94 |
|  | Monsoon (JJA) | 211 | 1.57 | 3.53 | 0.89 |
|  | Post Monsoon (SON) | 410 | 7.23 | 16.06 | 0.5 |

| | | | | | |
|---|---|---|---|---|---|
| | Winter (DJF) | 512 | -1.09 | 2 | 0.97 |
| JBPR | Pre Monsoon (MAM) | 276 | 1.49 | 3.48 | 0.86 |
| | Monsoon (JJA) | 160 | 0.97 | 2.8 | 0.9 |
| | Post Monsoon (SON) | 507 | -2.52 | 3.89 | 0.98 |
| | Winter (DJF) | 540 | -1.72 | 2.5 | 0.96 |
| JIPR | Pre Monsoon (MAM) | 276 | 3.67 | 8.28 | 0.16 |
| | Monsoon (JJA) | 160 | 2.28 | 7.53 | 0.73 |
| | Post Monsoon (SON) | 507 | -0.47 | 8.05 | 0.88 |
| | Winter (DJF) | 540 | -0.05 | 5.4 | 0.58 |
| JPGI | Pre Monsoon (MAM) | 662 | 0.69 | 4.15 | 0.93 |
| | Monsoon (JJA) | 188 | -2.79 | 4.41 | 0.8 |
| | Post Monsoon (SON) | 644 | -1.58 | 4.32 | 0.97 |
| | Winter (DJF) | 674 | -0.57 | 2.63 | 0.87 |
| PUNE | Pre Monsoon (MAM) | 456 | -7.28 | 8.21 | 0.92 |
| | Monsoon (JJA) | 212 | -7.06 | 8.02 | 0.81 |
| | Post Monsoon (SON) | 424 | -6.32 | 7.14 | 0.94 |
| | Winter (DJF) | 73 | -4.1 | 4.65 | 0.94 |
| RIPR | Pre Monsoon (MAM) | 573 | -0.98 | 3.59 | 0.94 |
| | Monsoon (JJA) | 135 | -1.94 | 3.53 | 0.74 |
| | Post Monsoon (SON) | 488 | -2.79 | 3.96 | 0.98 |
| | Winter (DJF) | 531 | -2.21 | 2.81 | 0.97 |
| KRKL | Pre Monsoon (MAM) | 711 | -1.28 | 3.37 | 0.97 |
| | Monsoon (JJA) | 225 | 0.52 | 2.94 | 0.8 |
| | Post Monsoon (SON) | 690 | -0.8 | 4.37 | 0.89 |
| | Winter (DJF) | 323 | -1.26 | 3.58 | 0.95 |
| KYKM | Pre Monsoon (MAM) | 647 | 0.61 | 3.44 | 0.94 |
| | Monsoon (JJA) | 212 | 0.03 | 3.01 | 0.87 |
| | Post Monsoon (SON) | 589 | 1.07 | 3.57 | 0.92 |
| | Winter (DJF) | 697 | -0.03 | 3.11 | 0.95 |
| MPTM | Pre Monsoon (MAM) | 632 | -0.28 | 3.26 | 0.94 |
| | Monsoon (JJA) | 223 | 0.96 | 3.31 | 0.8 |
| | Post Monsoon (SON) | 655 | -2.26 | 4.27 | 0.96 |
| | Winter (DJF) | 419 | -2.55 | 3.52 | 0.96 |
| DWRK | Pre Monsoon (MAM) | 597 | -1.02 | 2.53 | 0.91 |
| | Monsoon (JJA) | 218 | 1.42 | 3.4 | 0.96 |
| | Post Monsoon (SON) | 614 | -0.92 | 3.8 | 0.95 |
| | Winter (DJF) | 665 | -1.43 | 2.77 | 0.91 |
| GOPR | Pre Monsoon (MAM) | 656 | -1.4 | 4.46 | 0.89 |
| | Monsoon (JJA) | 231 | 2.1 | 3.65 | 0.8 |
| | Post Monsoon (SON) | 318 | 1.42 | 3.35 | 0.96 |
| | Winter (DJF) | 420 | -1.64 | 2.78 | 0.92 |

| | | | | | |
|---|---|---|---|---|---|
| PNJM | Pre Monsoon (MAM) | 398 | 3.6 | 7.88 | 0.74 |
| | Monsoon (JJA) | 75 | 3.57 | 11.41 | 0.38 |
| | Post Monsoon (SON) | 277 | 0.01 | 4.23 | 0.86 |
| | Winter (DJF) | 0 | NaN | NaN | NaN |
| TRVM | Pre Monsoon (MAM) | 631 | -2.26 | 4.7 | 0.9 |
| | Monsoon (JJA) | 199 | -0.51 | 2.3 | 0.92 |
| | Post Monsoon (SON) | 617 | -1.17 | 3.85 | 0.89 |
| | Winter (DJF) | 626 | -2.74 | 4.84 | 0.89 |
| BWNR | Pre Monsoon (MAM) | 644 | 13.88 | 16.5 | 0.29 |
| | Monsoon (JJA) | 0 | NaN | NaN | NaN |
| | Post Monsoon (SON) | 0 | NaN | NaN | NaN |
| | Winter (DJF) | 596 | 0.6 | 9.48 | 0.16 |
| SGGN | Pre Monsoon (MAM) | 680 | -0.85 | 2.76 | 0.93 |
| | Monsoon (JJA) | 192 | -0.84 | 4.57 | 0.94 |
| | Post Monsoon (SON) | 712 | -2.51 | 4.04 | 0.97 |
| | Winter (DJF) | 690 | -2.05 | 2.67 | 0.95 |


## 3. Results and discussion

### 3.1 Inter-comparison of INSAT-3DR and Indian GNSS IPWV

From  Figure 3, The Taylor diagram to evaluate the skill characteristics of the annual distribution of IPWV retrieved from INSAT-3DR satellite with 19 GNSS IPWV at different geographical locations (Figure 2) over Indian subcontinent during the period of 1 January 2017 to 30 June 2018. Further tailor diagram displaying three statically skill metrics: distribution of the correlation coefficient, root mean square error (RMSE) and standard deviation. If an IPWV performs nearly perfectly, its position in the diagram is expected to be very close to the observed point (Figure 3). An attempt have been made to evaluate the IPWV retrieved from INSAT-3DR satellite with GNSS observations show the root mean square error (RMSE) of 8 inland stations out of 10 stations lies between 4 to 6 mm except 8 mm and 12 mm for Jalpaiguri (JPGI) and Dibrugarh (DBGH) stations respectively. The observation points in case of Dibrugarh (DBGH) are more symmetrical (or association) than Jalpaiguri (JPGI) even RMSE values are higher (Figure 4).The value of Correlation Coefficient (CC) and bias for inland stations lie in the range (0.72 to 0.93) & (-3.0 mm to +3.0 mm) respectively. Similarly, for all the coastal stations the value of CC and bias lie in the range (0.67 to 0.88) & (-3.0 mm to +3.0 mm) respectively. RMSE for 7 coastal stations out of 8 stations lie between 5 mm to 7 mm except 9 mm of Panjim. The value of CC and bias and RMSE for desert station (SGGN) 0.88, -1.4 mm and 4.42 mm respectively (Table 3).

The correlation coefficient of IPWV varies from 0.60 to 0.89 of all the stations for the pre monsoon season. IPWV retrieved from INSAT-3DR satellite with respect to GNSS IPWV are having the negative biases ranges (-6.7 mm to -0.39 mm) which are indicating underestimation of IPWV  at the stations of ARGD, DBGH, DELH, NGPR, JIPR, JPGI, RIPR, GOPR, PNJM, TRVM &

SGGN. The stations JBPR, PUNE, KRKL, KYKM, MPTM, DWRK, and BWNR are having the
positive biases ranges (0.03 to 2.54 mm) which are indicating overestimation of IPWV by INSAT-
3DR during pre-monsoon season. RMSE ranges between 3.5 mm to 10 mm (Table 4).
The correlation coefficient of IPWV varies from 0.60 to 0.90 of all the stations during monsoon
season except TRVM (0.1), KYKM (0.31) and KRKL (0.15) respectively. The stations  ARGD,
DBGH, DELH, JBPR, JIPR, JPGI, PUNE, KRKL, KYKM, GOPR, BWNR, PNJM, TRVM and
SGGN are having the negative biases ranges (-0.39 mm to -12.39 mm)  which are indicating the
underestimation of IPWV by INSAT-3DR as compared to MPTM, NGPR & BHPL are having the
positive biases ranges of (0.39 mm to 2.47 mm) during monsoon season. RMSE ranges of 4.23
mm to 14.71 mm (Table 4).
The correlation coefficient of IPWV varies from 0.60 to 0.98 of all the stations during post
monsoon season except TRVM (0.42), PNJM (0.2), MPTM (0.48), KYKM (0.50) and DBGH (-
0.28) respectively. The stations DBGH, DELH, KRKL, MPTM, PNJM, TRVM and SGGN are
having the negative biases ranges (-0.32 mm to -6.10 mm) except DBGH (-22.52 mm) which are
indicating the underestimation of IPWV by INSAT-3DR as compared to ARGD, BHPL, NGPR,
JBPR, JIPR, JPGI, PUNE, RIPR, KYKM, GOPR, DWRK, BWNR are having the positive biases
ranges of (0.88 mm to 9.43 mm) during post-monsoon season. RMSE ranges from 3.94 mm to
13.49 mm except PNJM (18.73 mm) & DBGH (27.74 mm) respectively (Table 4).
The correlation coefficient of IPWV varies from 0.64 to 0.95 of all the stations during winter
season except DBGH (0.48), JPGI (0.50) respectively. The stations BHPL, DBGH NGPR, JBPR,
JIPR, JPGI, PUNE, RIPR, KRKL, KYKM, MPTM, GOPR, DWRK, PNJM, TRVM, BWNR &
SGGN are having the positive biases ranges (0.61mm to 5.90) which are indicating the
overestimation of IPWV by INSAT-3DR as compared to ARGD (-0.84 mm) & DELH (-1.51mm)
during winter season. RMSE ranges of 2.99 mm to 8.53mm (Table 4).
Scatter plot of hourly INSAT-3DR IPWV and GNSS IPWV plotted in Figure 4 using hexagonal
binning. The number of occurrences in each bin is colour-coded (not on a linear scale). It is now
possible to see where most of the data lie and a better indication of the relationship between GNSS
IPWV and INSAT-3DR IPWV are revealed.
Stations TRVM, KYKM, KRKL, PNJM, MPTM, JPGI and DBGH are poorly correlated (INSAT-
3DR vs. GNSS) averaging of INSAT-3DR pixels in gridded data contains both sea and
mountainous land together along with topographically diverse terrains around these stations.
Similar behavior is also seen in annual analysis of IPWV in coastal stations with the above said
reasons.
It is seen that discrepancies arise because the wet mapping functions that used to map the wet delay
at any angle to the zenith do not represent the localized atmospheric condition particularly for
Narrow towering thunder clouds and non-availability of GPS satellites in the zenith direction
(Puviarasan et al., 2020).
Large or small bias between IPWV retrieved from INSAT-3DR and GNSS exists due to
limitations of the INSAT-3DR retrievals and calibration uncertainties in the radiance measured by
INSAT-3DR. Another possibility of operation differences in IPWV measurements adopted in
GNSS /INSAT-3DR in respect to mapping functions /weighting functions.
The results indicate that the RMSE values increases significantly under the wet conditions (Pre
Monsoon & Monsoon season) than under dry conditions (Post Monsoon & winter season) (Table
4).The study showed differences in the magnitude and sign of bias of INSAT-3DR with respect to
GNSS IPWV from station to station and season to season. The data quality of INSAT-3DR IPWV
may be improved due to proper bias correction coefficient applied before physical retrievals of
IPWV during clear sky pixels.

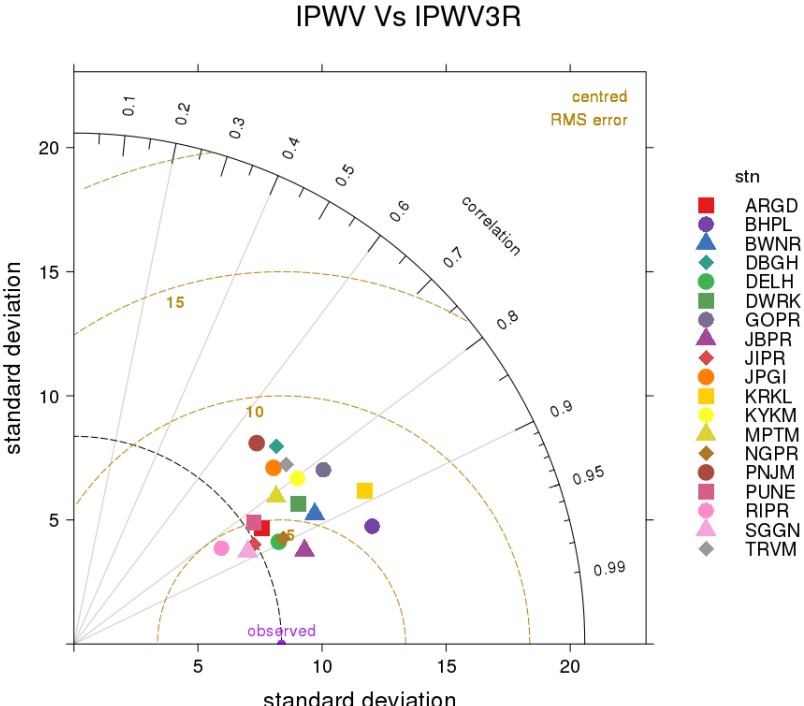



Figure 3. Taylor diagram of INSAT-3DR vs. Indian GNSS retrievals.

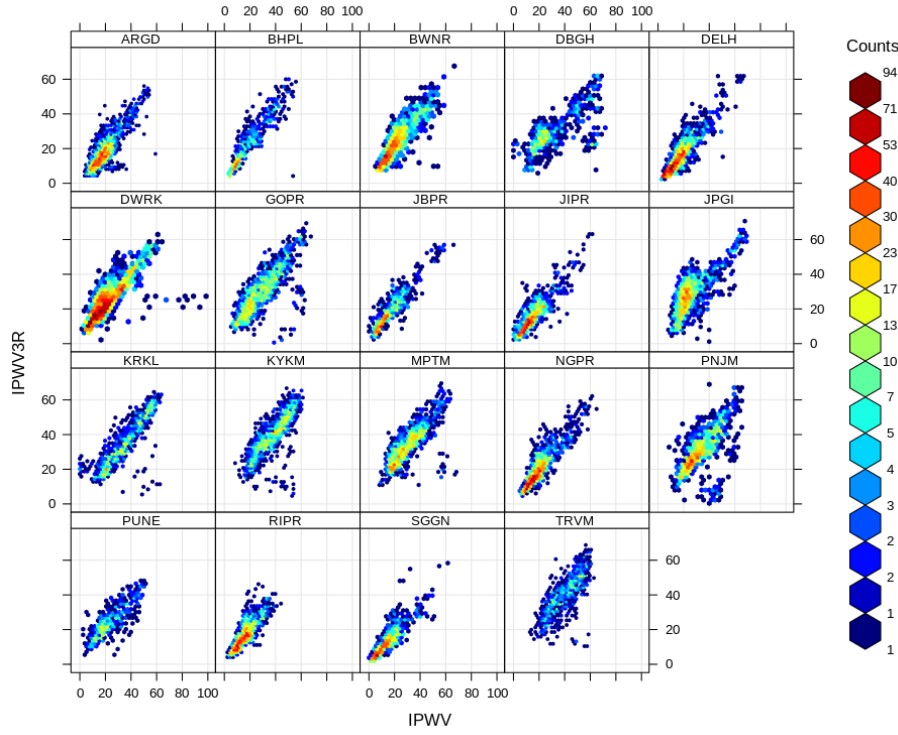


Figure 4. Scatter plot of hourly INSAT-3DR IPWV vs. GNSS IPWV using hexagonal binning.

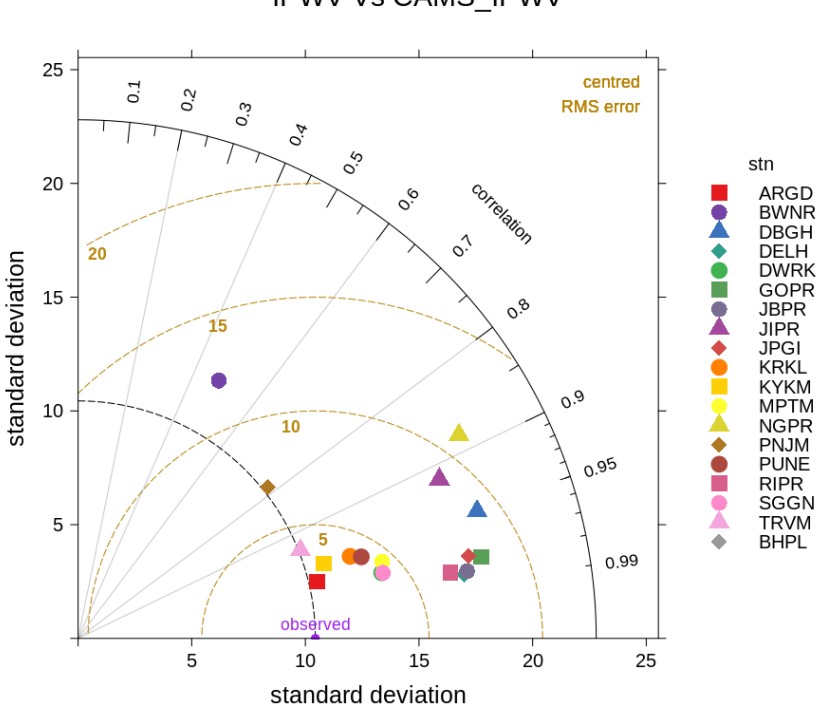


Figure 5.Taylor diagram of CAMS vs. Indian GNSS retrievals.

## 3.2 Inter-comparison of CAMS reanalysis and Indian GNSS IPWV

From the Figure 5, the Taylor diagram evaluates the skill characteristics in terms of RMSE, Correlation Coefficient and Standard Deviation of the annual distribution of IPWV retrieved from CAMS  with 19 GNSS IPWV at different geographical locations (Figure 5) over Indian subcontinent during the period of 1 January 2018 to 31 December 2018. The root mean square error (RMSE) between CAMS reanalysis & GNSS data retrievals of 9 inland stations out of 10 stations lies between 3 to 7 mm except 9 mm for Nagpur (NGPR) station respectively. The value of Correlation Coefficient (CC) and bias for inland stations lie in the range (0.88 to 0.99) & (-3.0 mm to +3.0 mm, except Pune, -6.69 mm) respectively (Table 5).

Root Mean Square Error (RMSE) for 7 coastal stations out of 8 stations lie between 3 to 7 mm except 14.0 mm of Bhubaneswar (BWNR). The value of CC and bias lie in the range (0.78 to 0.98 except 0.48 BWNR) & (-2.0 mm to +2.0 mm except +7.5 mm at BWNR) respectively. The value of CC and bias for desert station (SGGN) 0.88 and -1.4 mm respectively. The desert station RMSE, CC & Bias are 3.37 mm, 0.98 and -1.74 mm respectively (Table 5).

The correlation coefficient of IPWV varies from 0.74 to 0.97 of all the stations except JIPR (0.16) & BWNR (0.29) for the pre monsoon season. IPWV retrieved from CAMS reanalysis with respect to GNSS IPWV are having the negative biases ranges (-7.28 mm to -0.28 mm) which are indicating underestimation of IPWV  at the stations of ARGD, DELH, NGPR, PUNE, RIPR, KRKL, MPTM, DWRK, GOPR, TRVM, SGGN. The stations DBGH, JBPR, JIPR, JPGI, KYKM, PNJM and BWNR are having the positive biases ranges (0.61 mm to 13.88 mm) which are indicating overestimation of IPWV by CAMS during pre-monsoon season. RMSE ranges between 2.27 mm to 8.28 mm except BWNR (16.50 mm) (Table 6).

The correlation coefficient of IPWV varies from 0.73 to 0.96 of all the stations during monsoon season except PNJM (0.38) respectively. The stations  ARJD, JPGI, PUNE, RIPR, TRVM and SGGN are having the negative biases ranges (-0.51 mm to -7.28 mm)  which are indicating the underestimation of IPWV by CAMS reanalysis as compared to DBGH, DELH, NGPR, JBPR, JIPR, KRKL, KYKM, MPTM, DWRK, GOPR & PNJM are having the positive biases ranges of (0.03 mm to 6.60 mm) during monsoon season. RMSE ranges from 2.30 mm to 11.41 mm. Data is not available at the stations of BHPL & BWNR (Table 6).

The correlation coefficient of IPWV varies from 0.86 to 0.99 of all the stations during post monsoon season except NGPR (0.50) respectively. The stations ARJD, DELH, JBPR, JIPR, JPGI, PUNE, RIPR, KRKL, MPTM, DWRK, TRVM, SGGN are having the negative biases ranges (-0.47 mm to -6.32 mm) which are indicating the underestimation of IPWV by CAMS reanalysis as compared to DBGH, NGPR, KYKM, GOPR, PNJM are having the positive biases ranges of (0.01 mm to 7.23 mm) during post-monsoon season. RMSE ranges from 3.35 mm to 8.05 mm except NGPR (16.06 mm) respectively (Table 6). During this transition time most parts of the Indian region remain gradually dry and decrease in water content as compared to the North East and

Southern parts of India. It has been observed in this analysis during post-monsoon season, stations located in dry/wet regions of India CAMS data under/over estimates with respect to GNSS.

The correlation coefficient of IPWV varies from 0.87 to 0.97 of all the stations during winter season except DBGH (0.49) JIPR (0.58) & BWNR (0.16) respectively. The stations ARJD, DBGH, DELH, NGPR, JBPR, JIPR, JPGI, PUNE, RIPR, KRKL, KYKM, MPTM, DWRK, GOPR, TRVM, SGGN are having the negative biases ranges (-0.03 mm to -4.10 mm) which are indicating the underestimation of IPWV by CAMS reanalysis as compared to BWNR are having the positive biases of (0.60 mm) during winter season. RMSE ranges of 1.74 mm to 9.48 mm respectively (Table 6).

During winter season over Indian region, local effects which play an important role moisture development are suppressed from their importance due to sparse observation network data and optimization of random and systematic errors which is further utilized for effective improvement in model predictions (Inness et al., 2019).

CAMS data used in this study have consistency and homogeneous spatial with reduced bias and better performance of model physics and dynamics due to assimilation of new data sets (Inness et al., 2019). But over Indian domains during pre-monsoon season land stations are mainly affected by local convective developments of shorter time scale of a few hours which is not captured by the CAMS data and a dry bias prevails in most of the stations mentioned above.

Few GNSS data is assimilated for Indian region in the latest CAMS Data sets. During monsoon season 6 stations mentioned above are underestimating IPWV with CAMS data due to complex and rugged topographic terrains which are not well captured in CAMS data due to very few observations are available in these locations. In almost all other stations IPWV values are overestimated as the global features of monsoon flow are well captured by the CAMS data. The similar findings (overestimate or underestimate) are also observed with GNSS data for above mentioned stations except PNJM and BWNR where the meteorological sensor gets replaced 2 to 3 times during the year of 2018. Standard deviation (SD) between CAMS reanalysis and Indian GNSS retrievals is more dispersed from their mean values (Figure 5).

**3.3 Inter-comparison of CAMS reanalysis and INSAT-3DR IPWV**

The correlation coefficient (CC) computed between INSAT-3DR and CAMS reanalysis, IPWV retrievals are negatively correlated in almost entire the land area, except pockets of Indo Gangetic Plain (IGP) of Indian region for winter months. The computed value of CC lies within the range 0.2 to -0.5 in the land area. Over Ocean retrievals the values of CC are slightly positive side (0.0 to 0.5) in the entire area of Bay of Bengal and Arabian Sea except off shore area on both east and west side in winter months (Figure 6). This poor resemblance between the results (INSAT-3DR and CAMS) may be due to the interpolated values of coarser resolution CAMS data.INSAT-3DR satellite based data have diverse, covariant information content, different temporal coverage and have smaller ability with respect to representative observations in CAMS.

In pre-monsoon season the value of CC between INSAT-3DR and CAMS reanalysis retrievals is
positive (0.0 to 0.6) over Oceanic entire areas of Bay of Bengal and Arabian Sea except few
patches in Arabian Sea. Over land the values are slightly positive (0.0 to 0.2) in many areas and
slightly negative (0.0 to -0.3) for pockets of the North West and Central India region (Figure 6).
During monsoon month the value of CC over land area are mostly positively correlated (0.0 to 0.7)
except the belt of monsoon trough and south India which have shown appreciably low value of CC
(-0.3 to -0.5). This might be due to the presence of clouds on both sides of monsoon trough and
southern belt of India during monsoon season. (Figure 6).
In post monsoon season months the value of CC between INSAT-3DR and CAMS reanalysis
retrievals are positive (0.0 to 0.7) for both land and oceanic areas almost entirely except some areas
of North of Bay and Bengal and South East Arabian Sea (Figure 6).
The differences in the magnitude and sign of CC of INSAT-3DR with respect to CAMS reanalysis
IPWV may be due to lack of assimilation of quality controlled data over Indian domain. This may
be due to limitations of the design of the instrument /sensor on board on INSAT-3DR or retrieval
algorithm of IPWV. Therefore, it will affect the overall collocations in matchup data sets.
During winter season, positive biases ranges (0.0 to 5.0 mm) observed between CAMS reanalysis
and INSAT-3DR IPWV which are indicating overestimation of CAMS IPWV over land and
oceanic region except east and west coast of India including Arabian Sea (12º N to 28º N), some
pockets of South East Bay of Bengal (BOB) and Himalayan region ranges (-2.5 mm to -5.0 mm)
which indicates underestimation of CAMS IPWV respectively (Figure 7).
During pre-monsoon season, positive biases ranges (0.0 to 10.0 mm) observed between CAMS
reanalysis and INSAT-3DR IPWV which indicates overestimation of CAMS IPWV over land and
oceanic region except some parts of North West of Arabian Sea and Himalayan region ranges (-
0.0 mm to -3.0 mm) which  indicates underestimation of CAMS IPWV respectively (Figure 7).
During monsoon season, positive biases ranges (2.5 to 10.0 mm) observed between CAMS
reanalysis and INSAT-3DR IPWV which indicates overestimation of CAMS IPWV over land and
oceanic region except Himalayan region ranges (-2.5 mm to -5.0 mm) which indicates
underestimation of CAMS IPWV respectively (Figure 7).
During post monsoon season, positive biases ranges (0.0 to 6.0 mm) observed between CAMS
reanalysis and INSAT-3DR IPWV which  indicates overestimation of CAMS IPWV over land and
oceanic region except Arabian Sea (19º N to 29º N) and Himalayan region ranges (-2.5 mm to -
6.0 mm) which indicates underestimation of CAMS IPWV respectively (Figure 7).

The IPWV retrieved from CAMS reanalysis overestimated with respect to INSAT-3DR IPWV
over land and oceanic region for all the seasons except Himalayan region and some parts of
Arabian Sea and BoB. This occurred because the infrared and microwave radiometer observations
of land and oceans had been assimilated into the model, which has the higher systematic humidity
when it was compared with Radiosonde data (Andersson et al., 2007). Underestimation of CAMS
IPWV compared with INSAT-3DR over Himalayan region may be due to presence of rugged
terrain/orographic features in the retrieval of IPWV.

RMSE values during winter season ranges (7.5 mm to 13.0 mm) over land region (20º N to 35º N)
and the entire Arabian Sea. Above 35º N latitude including Himalayan region, RMSE values are
less than 7.5 mm. RMSE values ranges (13 mm to 20 mm) observed over the Southern peninsula
of India and BoB region respectively (Figure 8).

RMSE values during pre-monsoon season ranges (2.5 mm to 13.0 mm) over land region (18º N to
40º N), Arabian Sea and Himalayan region observed. RMSE values ranges (13 mm to 20 mm) are
over the Southern peninsula of India, Indo Gangetic Plains (IGP) and BoB region respectively
(Figure 8).

RMSE values during monsoon season ranges (14. mm to 20.0 mm) over land region (20º N to 35º
N) including North West of Arabian Sea and North East of BoB. Above 35º N latitude, South West
& South East of Arabian Sea including South East of BoB and Himalayan region RMSE values
are less than 8.0 mm respectively (Figure 8).

RMSE values during post-monsoon season less than 7.5 mm observed over land region including
both Arabian Sea as well as BoB region except Indo Gangetic Plains (IGP) and north East of BoB
ranges (13 mm to 17 mm) respectively (Figure 8).

Seasonal RMSE between CAMS reanalysis and INSAT-3DR (CAMS-INSAT) retrievals are
higher (>15 mm) over Bay of Bengal and pockets of Indo Gangetic Plains (IGP), North East (NE)
India, Southern Parts of India,  North Indian Ocean and Arabian Sea during pre-monsoon,
monsoon, post monsoon season and (< 15 mm) during winter season. Higher values of RMSE
prevails over the regions of higher moisture availability or water content in the Atmosphere.
(Figure 8).

**3.4 Distribution and Variability of IPWV retrieved from INSAT-3DR and CAMS reanalysis**

The annual mean value and standard deviation of both the retrievals INSAT -3DR sounder and
CAMS reanalysis data sets are presented in Figure 9.  The standard deviations of CAMS reanalysis
retrieval data set are appreciably high (0.0 to 14 mm) in both land and ocean areas as compared to
INSAT-3DR retrievals. This variation of higher spread from mean values may be due to the drier
bias present in the CAMS reanalysis data sets (Inness et al, 2019) with coarser resolution as
compared to INSAT-3DR retrievals.
The mean IPWV values vary in the range of 0–50 mm depending upon the region and prevailing
weather system affected throughout the year. Larger mean IPWVs occur in the coastal regions of
Indian Ocean regions compared to inland and desert regions due to warm air conditions as
compared to inland and ocean. The south foothill of Himalayas has the largest IPWV variation
with a SD ~16 mm (Figure 9). This is attributed to the monsoon season that results in large changes
in precipitation at different seasons in these regions. The seasonal distribution of mean IPWV and
standard deviation of CAMS and INSAT-3DR for monsoon and post monsoon increased in CAMS
data as compared to INSAT -3DR retrievals due to wet bias present in the CAMS data sets (Figure
552 10).
Over the oceanic region, seasonal mean IPWV of INSAT-3DR and CAMS  ranges from 25-40
mm (with standard deviation 6-15 mm) and 20-45 mm (SD 6-16 mm) and less than 25 mm with
SD of less than 6 mm for both INSAT-3DR and CAMS  IPWV over land region during winter
season  respectively (Figure 10).

Over the oceanic region, seasonal mean IPWV of INSAT-3DR and CAMS ranges from 30-45 mm
(with standard deviation 7-12 mm) and 35-55 mm (SD 10-16 mm). Over land region, seasonal
mean IPWV of INSAT-3DR and CAMS data ranges from 15-38 mm with SD of 2-10 and 20-40
mm with SD of 5-12mm during pre-monsoon season respectively (Figure 10).
Seasonal mean IPWV of INSAT-3DR ranges from 30 mm to more than 60 mm with SD of 2-14
mm and from 50 mm to more than 60 mm with SD of 4-16 mm of CAMS IPWV observed for both
land and oceans region during monsoon season respectively (Figure 10).

Over the oceanic region, seasonal mean IPWV of INSAT-3DR and CAMS  ranges from 35-55
mm (with standard deviation 6-10 mm) and 38-55 mm (SD 6-14 mm) and  over land region mean
IPWV of INSAT-3DR and CAMS data ranges from 15-35 mm with SD of 5-12 and 20-40 mm
with SD of 10-16 mm during post-monsoon season  respectively (Figure 10).

The Standard deviations values are higher over ocean as compared to land areas in every season
except post monsoon season (Figure 10).




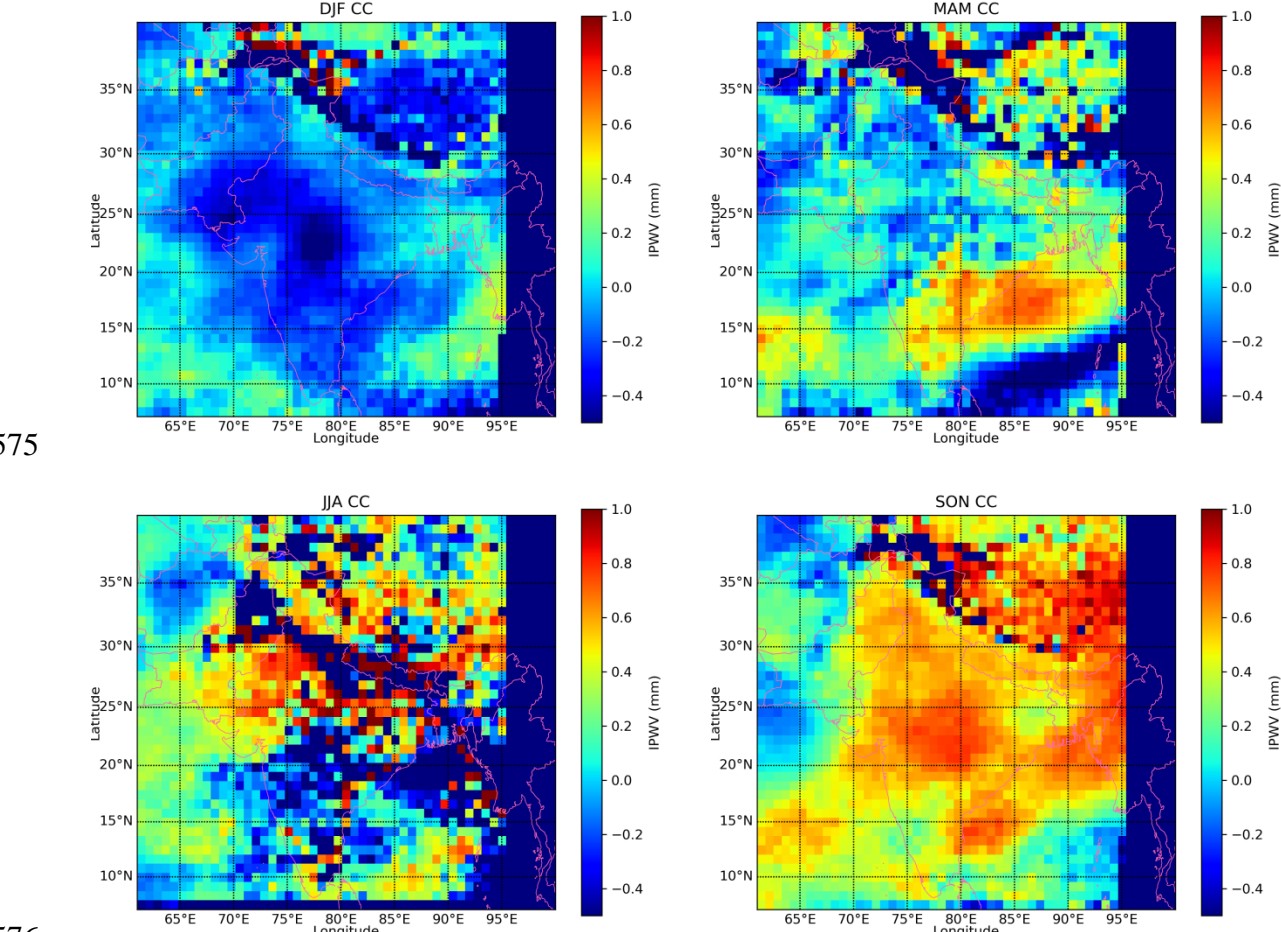

Figure 6. Seasonal Correlation Coefficient of CAMS and INSAT-3DR data



Figure 7. Seasonal bias of IPWV between CAMS and INSAT-3DR


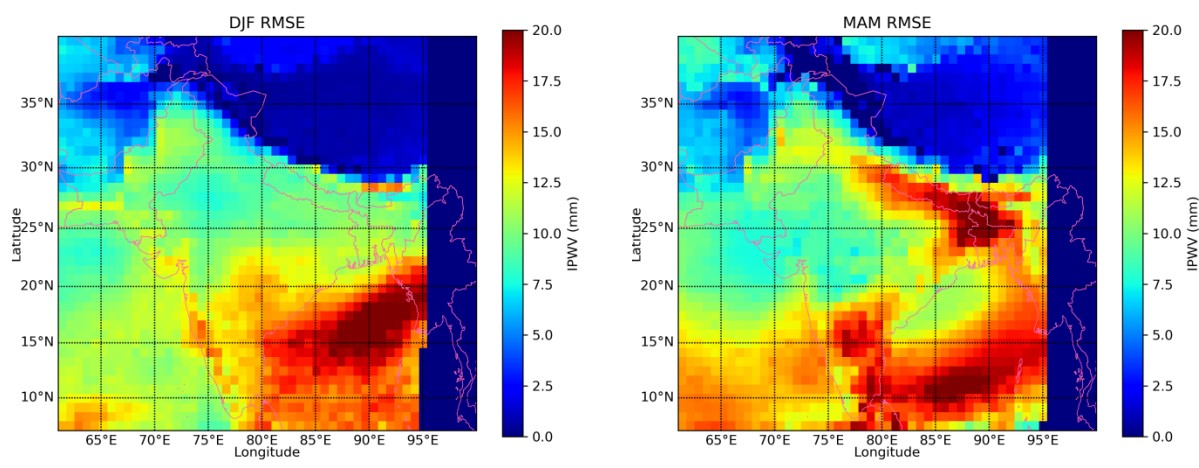


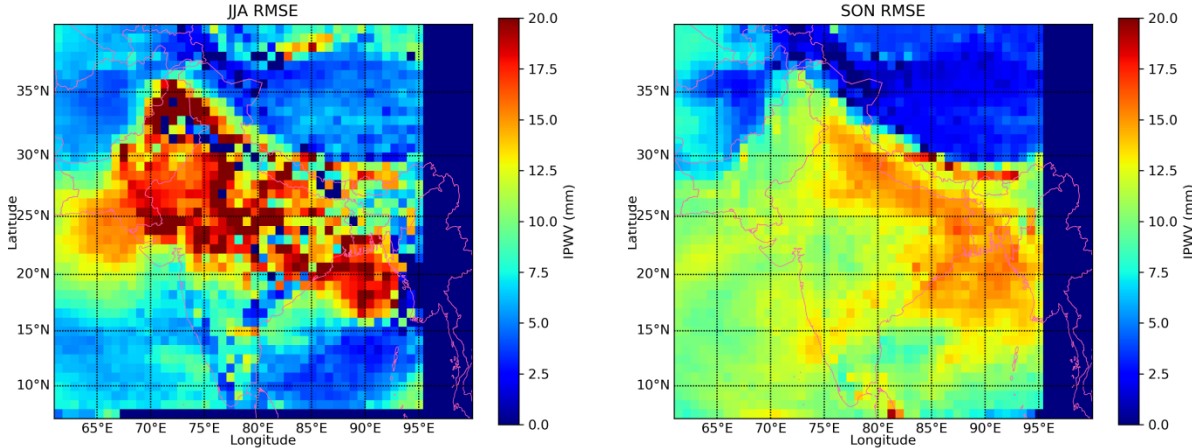


Figure 8. Seasonal RMSE between CAMS and INSAT-3DR

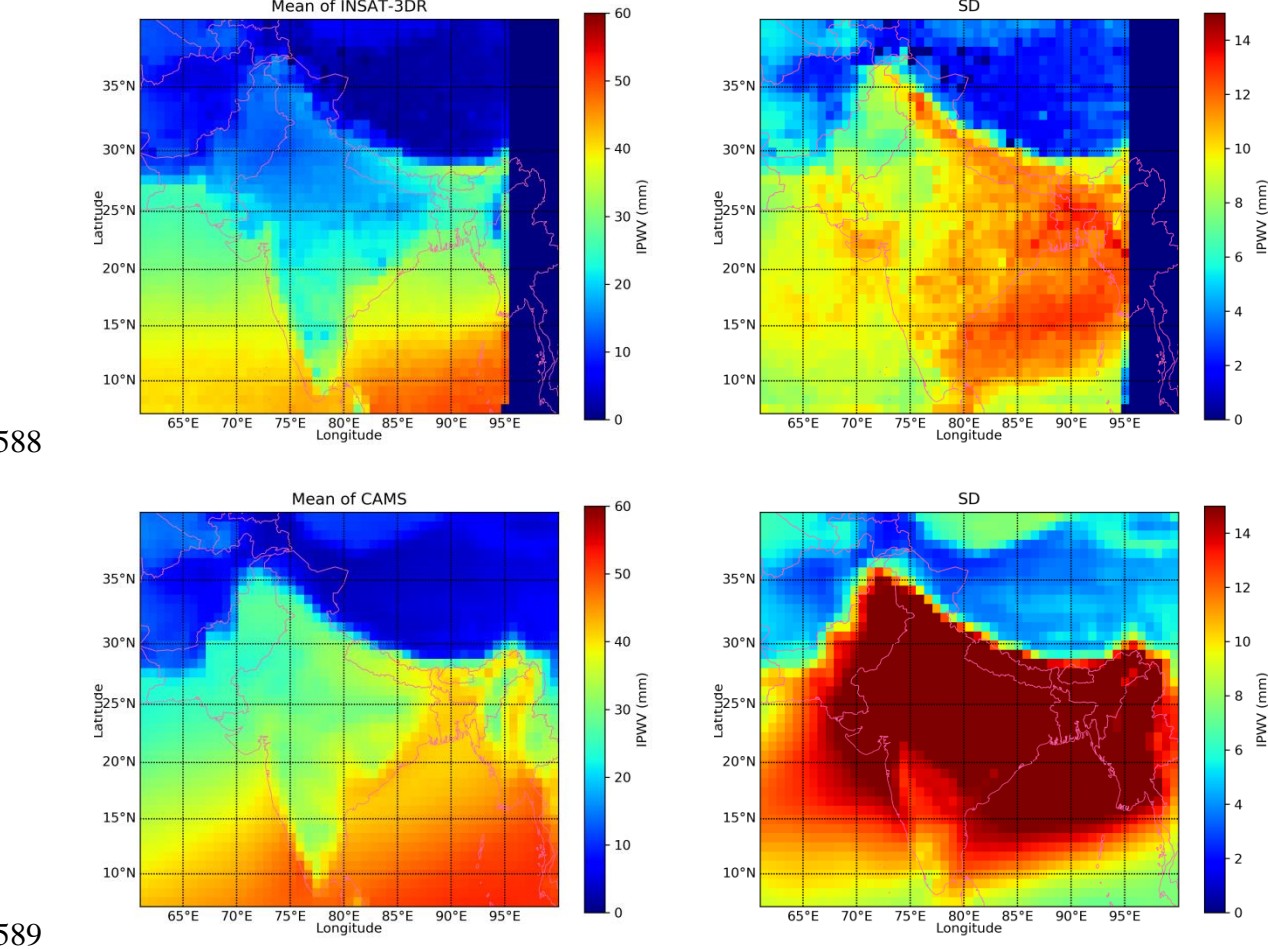



Figure 9. Means and SD of INSAT-3DR and CAMS IPWV for the year 2018

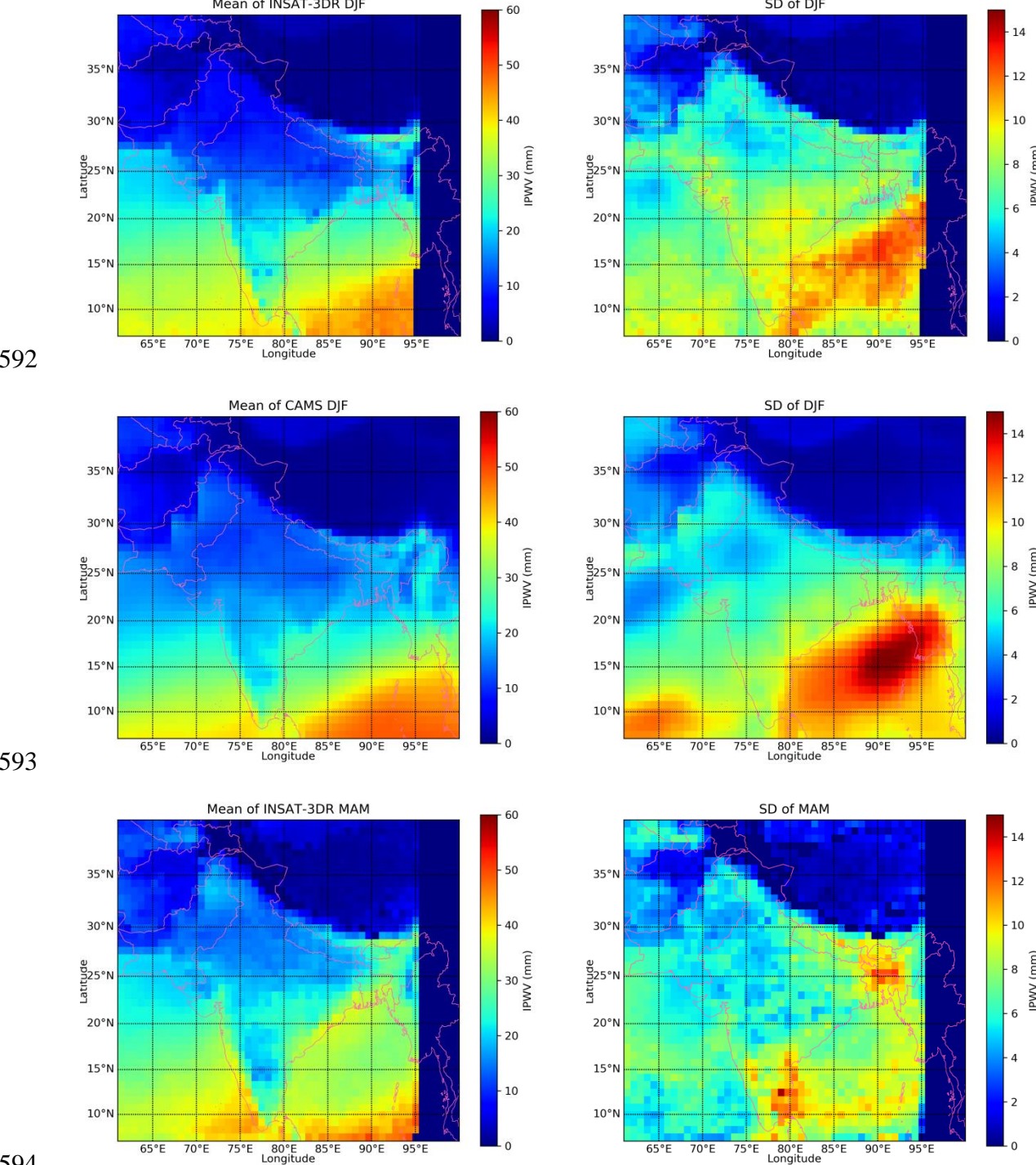





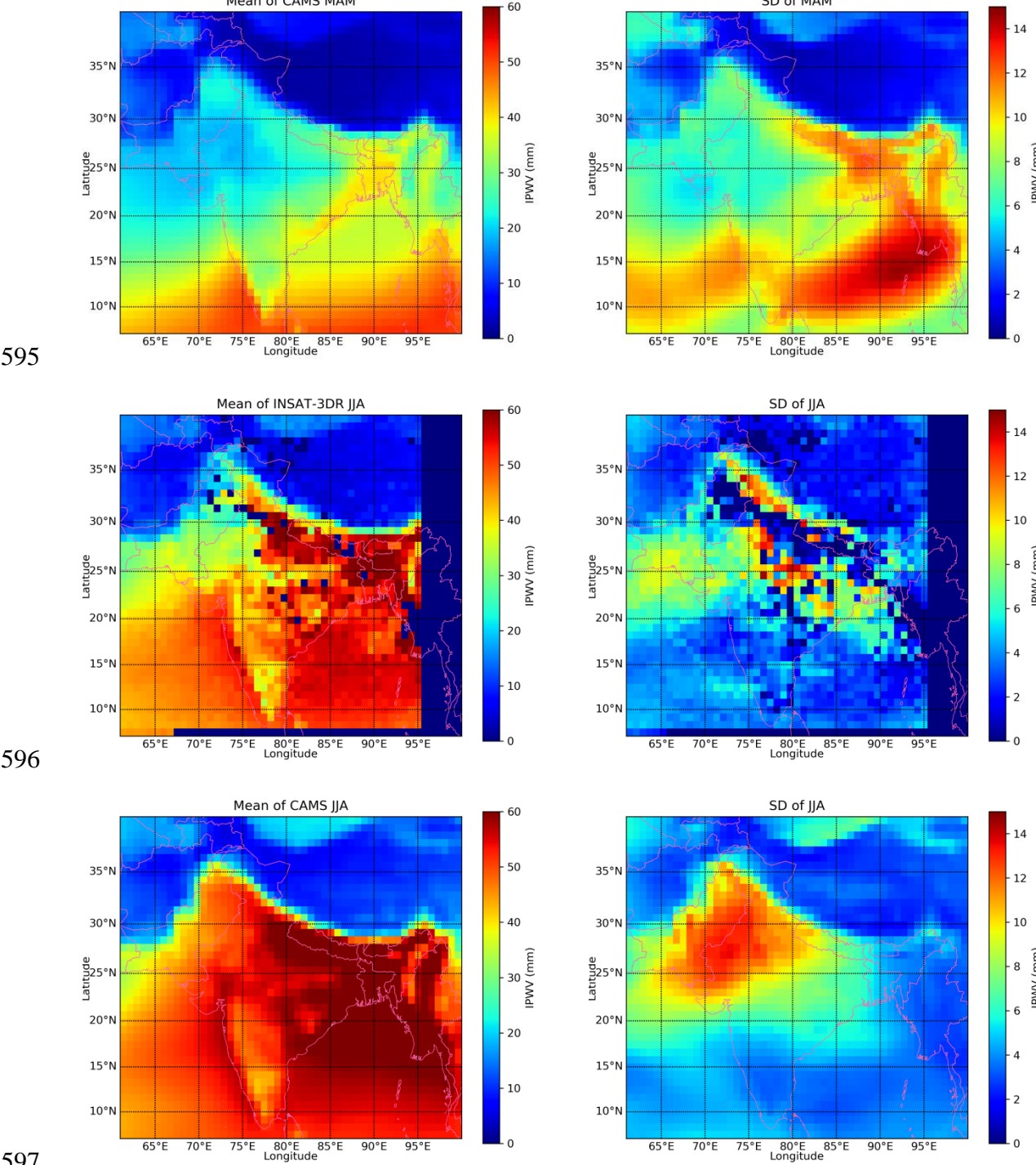




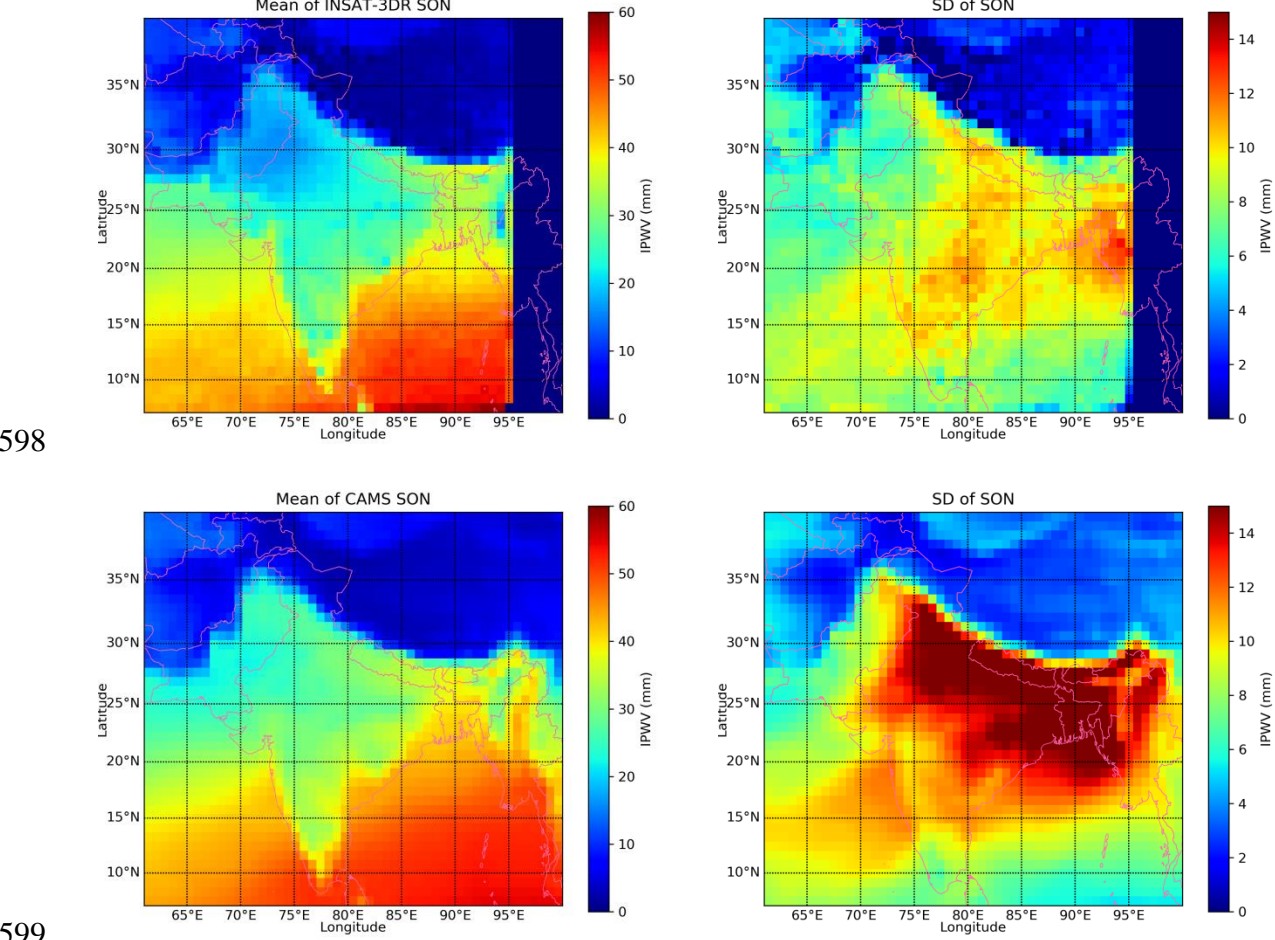


Figure 10. Seasonal Means and SDs of INSAT-3DR and CAMS retrieved IPWV for the year
2018

**4. Conclusions**

1. It is noticed that seasonal correlation coefficient (CC) values between INSAT-3DR and Indian GNSS data mainly lie within the range of 0.50 to 0.98 for all the selected 19 stations except Thiruvanathpuram (0.1), Kanyakumari (0.31), Karaikal (0.15) during monsoon and Panjim (0.2) during post monsoon season respectively. The seasonal CC values between CAMS and INSAT-3DR IPWV are ranges 0.73 to .99 except Jaipur (0.16) & Bhubneshwar (0.29) during pre-monsoon season, Panjim (0.38) during monsoon, Nagpur (0.50) during post-monsoon and Dibrugarh (0.49) Jaipur (0.58) & Bhubaneswar (0.16) during winter season respectively.

2. The RMSE values increases significantly under the wet conditions (Pre Monsoon & Monsoon season) than under dry conditions (Post Monsoon & winter season) and the differences in magnitude and sign of bias of INSAT-3DR, CAMS with respect to GNSS IPWV from station to station and season to season.

3. Large scale features of moisture flow are generally captured in CAMS reanalysis data
except localized features due to sparseness or very few numbers of the quality controlled
both ground as well as satellite data sets assimilated in the CAMS data over Indian region.
4. The differences in the magnitude and sign of CC of INSAT-3DR with respect to CAMS
reanalysis IPWV may be due to lack of assimilation of quality controlled data over Indian
domain. This may be due to limitations of the design of the instrument /sensor on board on
INSAT-3DR or retrieval algorithm of IPWV. Therefore, it will affect the overall
collocations in matchup data sets.
5. The IPWV retrieved from CAMS reanalysis overestimated with respect to INSAT-3DR
IPWV over land and oceanic region for all the seasons except Himalayan region and some
parts of Arabian Sea and BoB. This occurred because the infrared and microwave
radiometer observations of land and oceans had been assimilated into the model, which has
the higher systematic humidity when it was compared with Radiosonde data (Andersson et
al., 2007). Underestimation of CAMS IPWV compared with INSAT-3DR over Himalayan
region may be due to presence of rugged terrain/orographic features in the retrieval of
IPWV.
6. Seasonal RMSE between CAMS reanalysis and INSAT-3DR (CAMS-INSAT) retrievals
are higher (>15 mm) over Bay of Bengal and pockets of Indo Gangetic Plains (IGP), North
East (NE) India, Southern Parts of India, North Indian Ocean and Arabian Sea during pre-
monsoon, monsoon, post monsoon season and (< 15 mm) during winter season. Higher
values of RMSE prevails over the regions of higher moisture availability or water content
in the Atmosphere.
7. The mean IPWV values vary in the range of 0–50 mm depending upon the region and
prevailing weather system affected throughout the year. Larger mean IPWVs occur in the
coastal regions of Indian Ocean regions compared to inland and desert regions due to warm
air conditions as compared to inland and ocean. The south foothill of Himalayas has the
largest PWV variation with a SD ~16 mm.

This study will help to improve understanding regarding representation of uncertainties associated
with land, coastal and desert locations in term of seasonal flow of IPWV which is an essential
integrated variable in forecasting applications.
**5. Acknowledgements:** Authors are grateful to Director General of Meteorology for providing
data and support to accomplish this work and also thankful to the CAMS global web site data
(https://ads.atmosphere.copernicus.eu) link for providing the data for the above study.

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
