# Peer review of "2 Inter-comparison Review of IPWV retrieved from INSAT-3DR Sounder, GNSS & CAMS 3 Reanalysis Data 4 Ramashray Yadav, Ram Kumar Giri and Virendra Singh 5 Satellite Meteorology Division, India Meteorological Department, Ministry of Earth Sciences 6 New Delhi-110003 7 Abstract: 8 The spatiote"

_Atmospheric Measurement Techniques, 2021_

## Author Response (AR1)

**Reply of referee comments#1**

We are thankful for the valuable suggestions /comments of the learned referee for the paper

Review of Inter-comparison of retrievals of Integrated Precipitable Water Vapour IPWV) made by  INSAT-3DR satellite-borne Infrared Radiometer Sounding and CAMS reanalysis data  with ground-based Indian GNSS data.  Ramashray Yadav et al.

**Point wise reply is given below:**

**General observations:**

The authors have made a good effort in the present study by evaluating inter-comparison of ground based GNSS, remote sensing by INSAT-3DR satellite and CAMS re-analysis model based observations. This type of study is very important for operational forecasting services especially tropical reason where most of the weather system development is convective in Nature and of course moisture development also affected global and local features (rugged terrains, plain, coastal, topography etc).

**RC1:** The authors are properly compiled the objective of the study in the manuscript and appropriate to publish in the journal. However, I have given few comments /suggestions to further improve the manuscript as follows:

**Response**: We agree with the general observations mentioned about the paper.

**RC1:** Title of the manuscript is too lengthy, possible to make short.

**Response:** We have revised the title of manuscript and made it short. As per the suggestion the revised title may be changed as "Inter-comparison Review of IPWV retrieved from INSAT-3DR

Sounder, GNSS & CAMS Reanalysis Data"

**RC1:** During South West Monsoon season the Thiruvananthapuram (TRVM) has plenty of moisture available and ITCZ remain active while seasonal correlation coefficient with INSAT-

3DR and GNSS is very low. Explain it and add appropriately in the text.

**Response:** The IPWV derived from INSAT-3DR is averaged over 30x30 Km which contains both sea and mountainous land together along with topographically diverse terrains pixels around the

Thiruvananthapuram (TRVM), being a coastal station while IPWV derived from GNSS is column

IPWV over the station. This is the reasons why these are poorly correlated at costal stations.

**RC1:** Why the author considered INSAT-3DR instead of INSAT 3D? Give reason or may be some important points about the difference between two satellites. So it makes the case to use of the

INSAT-3DR data.

**Response:** The sounder payload of INSAT-3D and INSAT-3DR satellite are exactly same in terms of specification. The sounder payload of INSAT-3D satellite reached end of life in the month of

May 2020 that's why INSAT-3DR sounder data are used in the present study.

**RC1:** "In this paper, CAMS & INSAT-3DR retrieval has been compared and statistically analyzed with GNSS data taking as reference". This is the paper objective only compare the two products from different sources? Mention the clear-cut objective and benefit of the study in last para of the introduction section.

**Response:** Necessary changes has been made as proposed (line-81-85).

**RC1:** Line 159: "The full aperture internal Black-body calibration is performed every 30 min or on **command based whenever**. This enables the derivation of vertical profiles of temperature and humidity". Explain it clearly the mechanism of calibration and correct the sentence appropriately.

How it will be useful in operational forecasting and present work.

**Response:** Mechanism of calibration and how it is useful in operational forecasting and present work has explained in manuscript (line-166-173 & 175-177).

**RC1:** Line 179: You have used Ground based GNSS data as base for comparison with INSAT-

3DR and CAMS data. But the GNSS based data also associated with errors and may behave differently over land, coastal and desert locations. Explain the possible sources of GNSS errors in your analysis after the sentence in the line 179.

**Response**: The other possible sources of error associated with GNSS data are mean temperature of atmosphere, dynamical pressure and isotropic errors. These errors will vary with location and time of observation. The same has been added in the revised manuscript (line-188-190).

**RC1:** Line 140: RMSE values for Jalpaiguri (JPGI) and Dibrugarh (DBGH) stations shown higher, is there any specific reason for this finding, is association of the data values is also behave same way?

**Response:** The observation points in case of Dibrugarh (DBGH) are more symmetrical (or association) than Jalpaiguri (JPGI) even RMSE values are higher (Figure 4).

**RC1:** Also please explicitly mention the importance of CAMS data in weather forecasting over

Indian region in the manuscript.

**Response:** CAMS data is capable to capture large scale features of moisture flow and used to predicts large scale weather events such as western disturbances, cyclonic storm, monitoring of monsoon and same added in manuscript (Line 218-220).

**RC1:** It is suggested for future INSAT-3DR sounder PWV data performance over ocean and AERONET, PWV data as ocean play an important role and contributing differently thorough out the year.

Response: Yes, we agree with referee suggestions.

**RC1:** Besides these I could see other numerous minor typos/English grammar errors. I am listing few of them here and check carefully in whole manuscript.

Line No12: it may be retrieval data at the end

Line No 15: Complete the sentence-------appropriately.

Line 344 to 346: provide gap in mm in whole text whenever necessary and frame the sentence properly. Change it throughout the manuscript.

Line 403: use everywhere the same notation

**Response:** Necessary changes has been made in manuscript as proposed by referee.

**Reply of Referee comments#2**

We are thankful for the valuable suggestions /comments of the learned referee for the paper Review of Inter-comparison of retrievals of Integrated Precipitable Water Vapour IPWV) made by  INSAT-3DR satellite-borne Infrared Radiometer Sounding and CAMS reanalysis data  with ground-based Indian GNSS data.  Ramashray Yadav et al.

**Point wise reply is given below:**

**General observations:**

This paper presents a validation task of two IPWV (integrated precipitable water vapour) products (from INSAT-3DR and CAMS) using as reference ground-based data at 19 Indian GNSS stations. The novelty of the study is not high, but the obtained results are interesting to know more about the satellite and reanalysis uncertainties and to try to improve them. In this sense, the paper fits with the scope of the journal and it should be published after some revisions. The manuscript is full of errors and typos, e.g., the format of citations varies in the text, the tables appear all together at the end of Section 2, while all the figures appear at the end of Section 3, making the reading difficult for the reader. The introduction must be improved, since it is not clearly motivating the purpose of the objectives of the paper. The objectives should be moved from Section 3 to the introduction.

**Response:** We agree with the general observations raised by the learned referee and manuscripts is modified appropriately as per suggestions (line-84-88).

**RC#2:** Here some minor comments:

Title: Could be shorter? There is a lack of parenthesis in IPWV too.

**Response:** We have revised the title of manuscript and made it short. As per the suggestion the revised title may be changed as "Inter-comparison Review of IPWV retrieved from INSAT-3DR

Sounder, GNSS & CAMS Reanalysis Data".

**RC#2:** L25: CASMS?

**Response:** Replaced with CAMS (line-25).

**RC#2:** L43, L51 and L84: IPWV has been defined before in Line 34.

**Response:** modified appropriately (line-34-37).

**RC#2:** L44: column

**Response:** modified as suggested.

**RC#2:** L77: the citation format (Perez-Ramirez, D. et al. 2014) is not appropriate.

**Response:** Modified as suggested (line-78).

**RC#2:** L84: Precipitable instead of perceptible.

**Response:** replaced with Precipitable (line-86).

**RC#2:** L107: If the reference value is the GNSS data, i.e. Mi, the MB should be calculated as  the mean of the Oi-Mi differences instead of Mi-Oi differences.

**Response:**  Replaced with $O_i - M_i$ (Line-113-117) in manuscript.

**RC#2:** L206: how this interpolation is done?

**Response:** We use nearest neighbor interpolation techniques to interpolate CAMS with GNSS

data. In this method we evaluate each station to determine the number of neighboring grid cells in

0.75 x 0.75 box that surround the GNSS station and contain at least one valid CAMS reanalysis data (line-236-242).

We are thankful for the valuable suggestions /comments of the learned referee for the paper Review of Inter-comparison of retrievals of Integrated Precipitable Water Vapour IPWV) made by INSAT-3DR satellite-borne Infrared Radiometer Sounding and CAMS reanalysis data with ground-based Indian GNSS data. Ramashray Yadav et al.

**Point wise reply is given below:**

**General observations:**

This paper entitled 'Inter-comparison of retrievals of Integrated Precipitable Water vapor (IPW) made by INSAT-3DR' satellite-borne Infrared Radiometer Sounding and CAMS reanalysis data with ground-based Indian GNNSS data' deals with the validation of INSAT-3DR and CAMS water vapor products using as reference GPS retrievals in India. To date there plenty of papers dealing with the validation of satellite and global reanalysis models IPW. But this paper is of interest to scientific community because INSAT-3DR is a geostationary satellite that allows continuous monitoring of IPW in Indian sub-continent. Also, the results presented here serve to validate CAMS reanalysis model. Having both INSAT-3DR and CAMS high precision data is of great importance for numerical weather predictions (NWP). Thus, I consider that the study is of interest and publishable in Atmospheric Measurement Techniques. However, I consider that the manuscript needs to be further improved before its final publication.

**MAJOR REVISIONS:**

**RC#1.** The authors remark in the introduction (Lines 73-76) and in the results sections the importance of evaluating INSAT-3DR and CAMS over Oceans. Obviously, they do not have GPS measurements in remote oceanic regions. However, Maritime Aerosol Network offers a publicly free database of IPWV over oceans that are unique for the validations of satellites and global models IPVW products. Including such data in your validations will provide a unique value to the manuscript. See the references Smirnov et al., (2004, 2011) and Perez-Ramirez et al., (2019).

**Response**:1. We agree with the learned referee concern of Maritime Aerosol Network data. Recently we have modified our INSAT-3DR scan strategy over oceanic region and definitely we will incorporate this data in our future strategy with our New INSAT-3DR data sets. We have added the reference suitably in the manuscripts and definitely incorporate in future studies.

**RC# 2.** The database used for the validation is short. Why not using more years? Or why not using AERONET data? Another possibility is to estimate IPWV from ground-based temperature and relative humidity in remote areas (see Falaiye et al., 2018).

**Response:** We fully agree with the referee suggestion. The Indian GNSS network is recently established and that is why the validation time is short. But we will definitely incorporate other possibilities as suggested of IPWV estimate in our future studies. The study carried out by Falaiye et al., 2018 is very important for considering the conventional data from long term observing stations of Indian domain along with the available model to establish the similar empirical relationship of getting the precipitable water vapour. This will also support to generate improved climatological mean especially over the remote regions.

**RC#3.** There is a systematic lack of appropriate references in all the text. Appropriate references are needed to fulfill quality standard in Atmospheric Measurement Techniques publication. Some of the most important are:

**Response:** We agree with the referee's suggestions and a brief discussion along with references regarding Satellite, Mosel and Ground based IPWV measurements have been added in the manuscript.

a. **RC#** No discussion of other satellites that provide IPWV in the introduction (e.g. MODIS, SCIAMACHY, GOME-2, AIRS)

**Response:** In Global Ozone Monitoring Experiment (GOME) and Scanning Imaging Absorption Spectrometer for Atmospheric CHartography (SCIAMACHY), both used the principle of differential optical absorption spectroscopy in red spectral range of IPWV retrieval (Beirle et al, 2018). Atmospheric Infrared sounder is a hyper spectral instrument which collects radiances in 2378 IR channels with wavelength ranging from 3.7 to 15.4 μm. Cloud cleared radiances of AIRS were utilized in the retrieval of column integrated water vapour which is contributed by a number of channels having different sensitivity towards water vapour. (Aumann et al., 2003).

Moderate Resolution Imaging Spectroradiometer (MODIS) utilized infrared algorithm employs ratios of water vapor absorbingchannels at 0.905, 0.936, and 0.940 μm with atmospheric window channels at 0.865 and 1.24 μm in estimated the precipitable water vapour (Kaufman and Gao, 1992).

The uncertainties in the retrieval of precipitable water vapor from satellites (like errors of calibration of channels, viewing geometry, radiative transfer in the forward models) are already addressed by previous studies (Ichoku et al., 2005 for MODIS. Noel et al., 2008 for GOME-2 and SCIAMACHY, Susskind et al., 2003, 2006 for AIRS). Wagner et al., 2006 studied GOME data for the period of 1996-2002 and reported globally and yearly averaged 2.8 ±0.8% increase of total column precipitable water (excluding the ENSO period).

b. **RC#** No discussion of other global reanalysis models (e.g. MERRA-2, CFSR)

**Response:** The retrievals from reanalysis data sets Modern-Era Retrospective analysis for Research and applications-2 (MERRA-2) Gelaro et al., 2017 , Climate Forecast System Reanalysis (CFSR)(Saha et al., 2010) Data Archive at https://rda.ucar.edu/pub/cfsr.html utilized 3d-var data assimilation techniques and well captured the interannual variations of precipitable water vapour in the south of the Central Asia (Jiang et al., 2019).The study carried out by Berrisford et al., 2011, found ERA interim data set is superior in quality than ERA 40 during the period 1989-2008.

c. **RC#** No discussion of other ground-based techniques used for validation of IPWV (e.g. radiosondes, AERONET sun-photometry and microwave radiometry).

**Response:** Ramashray et al., 2020 carried out the validation of Indian GNSS IPWV with GPS Sonde data for the period of June 2017 to May 2018 over Indian region and found reasonably well in agreement with in situ observations. In situ Radiosonde observations generally suffer spatiotemporal inhomogeneity errors and differences in relative humidity measured by different sensors. In this study he brought out positive bias less than 4.0 mm for 7 stations, correlation coefficient greater than 0.85 and RMSE less than 5.0 mm for all 09 collocated GPS sonde stations. In this direction the work carried out by Turner et al., 2003, 5 % dry bias with Microwave Radiometer and Vaisala RS80-H will be very useful while dealing with such Radiosonde observations. Miloshevich et al 2009, found a similar limitation of Relative Humidity measurement with Vaisala RS92 Radio sonde and derived an empirical correction to remove the mean bias error, yielding bias uncertainty is independent of height.

The work done in the past by Smirnov et al., 2004, 2011,  in retrieving the precipitable water vapour from aerosol network data especially for marine areas is very helpful to carry out further studies in future with INSAT-3DR satellite observations over oceanic areas.

Validation with other ground based techniques Referee decision is well taken and will be carried out in future with longer duration and more number of GNSS stations.

d.   **RC#** No references to INSAT-3DR neither for instrument specifications nor for retrieval algorithm. Are data publicly available?

**Response (d):** ATBD of INSAT reference is added.

e.   **RC#** No references for GNSS network and/or data. Are data publicly available?

**Response:** Data supply Portal of INSAT as well as GNSS data is under final phase of its development and will be available for public soon. The data will be downloaded as per the data policy.

f.   **RC#** No references for CAMS model. The link where data were obtained is necessary.

**Response:** CAMS model reference is added.

g.   **RC#** No comparisons of the results with other obtained in previous studies.

**Response**: We respect the encouragement and suggestions made by the referee in exploring the scope of the study. The reference of comparison study of GNSS with Radiosone data has been added.

**MINOR REVISONS**

**RC#** Introduction section needs to be further improved and appropriately referenced.

**Response:** Modified as per suggestions.

**RC#** Line 37: Currently, remote sensing instrument cost has been reduced. Please rearrange.

**Response:** modified as per suggestion.

**RC#** Line 38: Give an appropriate discussion of remote sensing techniques with appropriate references.

**Response:** modified as per suggestion.

**RC#** Line 43: IPWV was already defined.

**Response:** modified as per suggestion.

**RC#** Lines 43-44: What do you mean 'surface radiation is completely absorbed by atmospheric water vapor in its way to the satellite'? Not all energy is absorbed. It depends on wavelength and water vapor content.

**Response:** Agreed with the referee suggestion and the same is modified in the manuscript.

**RC#** Lines 50-52: What are the advantages/disadvantages of geo-stationary satellites versus polar orbiting satellites? You need to discuss previous achievements by polar orbiting satellites.

**Response:** Geo satellites have higher temporal resolution and continuous coverage and are important for monitoring the extreme weather events. Polar satellites have higher advantage higher spatial resolution and can operate both cloudy and non-cloudy conditions more effectively as compared to Geo satellites. Courcoux and Schroder et al., 2013, worked out the accuracies of

Satellite Application Facility on Climate Monitoring (CMSAF) satellite Advanced Television and

Infrared Observation Satellite Operational Vertical Sounder (ATOVS) precipitable water vapour of about 2-4 mm with respect to radiosonde and Atmospheric Infrared Sounder (AIRS) data both over land and ocean with resolution 0.5 x 0.5.

**RC#** Line 66: What do you mean 'much improved biases'?

**Response:** Statement is corrected.

**RC#** Line 67: there is a typo in the references.

**Response:** Modified as suggested.

**RC#** Lines 73-76: Discussion about water vapor in oceanic areas need to be further improved.

See Perez-Ramirez et al., (2019).

**Response**: The study Perez-Ramirez et al., (2019) clearly brought out the importance of Maritime

Aerosol. Network (MAN) in retrieving the precipitable water vapour over remote oceanic areas.

The reanalysis model estimates have very good agreement with MAN with mean differences of ~

5 % and standard deviation of ~15 % under clear sky conditions.

we agree with the referee suggestion and reference of the same is added suitably.

**RC#** Methodology section is not well structured:

▪ Start with instrument and models (GNNS network, INSAT-3DR and CAMS). IPWV

mathematical definition (Line 143) must be in the first instrument you talk about (e.g. in the GNNS

network description).

▪ later continue with the description of statistic parameters.

▪ Finish the section with the matchups.

**Response:** Modified as per suggestions.

**RC#** Lines 94-95: It is unnecessary the information about the software you used for statistics.

**Response:**Software information has been removed from the manuscript.

**RC#** Line 123: NWP acronym has not been defined.

**Response:**NWP acronym has been mentioned in text.

**RC#** Section 2.3 Scan strategy of INSAT-3DR sounder: There are no references, so it seems that is the first time that is presented. Is there any literature about that? If so the section is unnecessary, just provide appropriate references.

**Response:**Reference (ATBD of INSAT) is added in the text.

**RC#** Lines 176-177: I do not understand the limitation of 5º.

**Response:**If we reduce the cut off angle from 5º multipath effect will occur and introduce inaccuracy in the IPWV estimation. Higher cut off angle ($> 5°$) may introduce dry bias in the

IPWV estimation and notable 0.8 mm error in IPWV (Emardson et al., 1998).

**RC#** Section 2.6: It is not clear how you do make the matchups between GNSS and CAMS. Also, in section 3.3 you perform an inter-comparison of CAMS with INSAT-3DR. How do you make these matchups?

**Response:**The CAMS reanalysis IPWV retrievals are interpolated to different geographical locations of 19 GNSS observations. We use nearest neighbor interpolation techniques to interpolate CAMS reanalysis with GNSS data. In this method we evaluate each station to determine the number of neighboring grid cells in 0.75° x 0.75° box that surround the GNSS station and contain at least one valid CAMS reanalysis data.

INSAT-3DR Data set has horizontal resolution at 30 x 30km ($3 \times 3$ pixels) for each cloud free pixel. Collocation match up has been created at 0.75° x 0.75° (about80 km) spatial resolution for comparison and performance of INSAT-3DR data with CAMS reanalysis data using bilinear interpolation technique.

**RC#** Table 1: There is typo in the units of central wavelengths.

**Response:**The units of central wavelengths added in the text (µm).

**RC#** Table 5 and Table 6. Please add to the legends that they are statistical analyses of the intercomparisons.

**Response:** Table 5 and Table 6 legends added that they are statistical analyses of the intercomparisons.

**RC#** Figure 4: Which data are you using in the Figure?

**Response:**INSAT-3DR and GNSS IPWV data are using in Figure 4.

**RC#** Lines 278-283: I do not understand the paragraphs. To me there is nothing related with the intercomparisons of IPWV?

**Response:**Paragraph has been removed from the manuscript.

**RC#** Lines 289-292: To me the influence of GPS error in the differences between GPS and satellites is negligible. Please quantify the error and improve the discussion. Differences in IPWV

must associated with the differences in the sampling area and with limitations in satellite retrievals.

**Response:**Yes, we also agree with this point and similar findings was observed in the study of

Puviarasan et al., 2020. But actual quantification of such type of errors we have not done, especially when the convective development is on other side of line of sight.

**RC#** Lines 293-296: Could satellite data be cloud-affected data?

**Response:**Satellite estimates are in cloud free regions.

**RC#** Lines 297-300: There is a miss of any proposal to improve data retrieval or data quality.

**Response:** The data quality of INSAT-3DR IPWV may be improved due to proper bias correction coefficient applied before physical retrievals of IPWV during clear sky pixels.

**RC#** Lines 348-351: Give references.
**Response:** Inness, A., Ades, M., Agustí-Panareda, A., Barré, J., Benedictow, A., Blechschmidt,

A.-M., Dominguez, J. J., Engelen, R., Eskes, H., Flemming, J., Huijnen, V., Jones, L., Kipling, Z.,

Massart, S., Parrington, M., Peuch, V.-H., Razinger, M., Remy, S., Schulz, M., and Suttie, M.: The

CAMS reanalysis of atmospheric composition, Atmos. Chem. Phys., 19, 3515–3556, https://doi.org/10.5194/acp-19-3515, 2019. (Earlier in reply of referee#3 comments Cohen et al., was added by mistake and now replaced by Innes et al., 2019)

**RC#** Lines 352-356: Give references

**Response:** Same as above.

**RC#** Section 3.3 Inter-comparison of CAMS reanalysis and INSAT-3DR IPWV: I suggest a plot with the differences to quickly visualize the inter-comparison.

**Response:** Plot of Seasonal bias (figure 7) may kindly be seen.

**RC#** Lines 389-391: Paragraph need to rearrange, I could not catch the main message

**Response:**The differences in the magnitude and sign of CC of INSAT-3DR with respect to CAMS

reanalysis IPWV due to lack of assimilation of quality controlled data over Indian domain. This may be due to limitations of the design of the instrument /sensor on board on INSAT-3DR or retrieval algorithm of IPWV. Therefore, it will affect the overall collocations in matchup data sets.

**RC#** There are lacks of discussions of Figure 7 and Figure 8 in the text.

**Response:** We agree with the comments.

**Seasonal Analysis:** During winter season, positive biases ranges (0.0 to 5.0 mm) observed
between CAMS reanalysis and INSAT-3DR IPWV which are indicating overestimation of CAMS
IPWV over land and oceanic region except east and west coast of India including Arabian Sea (12º
N to 28º N), some pockets of South East Bay of Bengal (BOB) and Himalayan region ranges (-2.5
mm to -5.0 mm) which indicates underestimation of CAMS IPWV respectively (Figure 7).
During pre-monsoon season, positive biases ranges (0.0 to 10.0 mm) observed between CAMS
reanalysis and INSAT-3DR IPWV which indicates overestimation of CAMS IPWV over land and
oceanic region except some parts of North West of Arabian Sea and Himalayan region ranges (-
0.0 mm to -3.0 mm) which indicates underestimation of CAMS IPWV respectively (Figure 7).
During monsoon season, positive biases ranges (2.5 to 10.0 mm) observed between CAMS
reanalysis and INSAT-3DR IPWV which indicates overestimation of CAMS IPWV over land and
oceanic region except Himalayan region ranges (-2.5 mm to -5.0 mm) which indicates
underestimation of CAMS IPWV respectively (Figure 7).
During post monsoon season, positive biases ranges (0.0 to 6.0 mm) observed between CAMS
reanalysis and INSAT-3DR IPWV which indicates overestimation of CAMS IPWV over land and
oceanic region except Arabian Sea (19º N to 29º N) and Himalayan region ranges (-2.5 mm to -
6.0 mm) which indicates underestimation of CAMS IPWV respectively (Figure 7).
The IPWV retrieved from CAMS reanalysis overestimated with respect to INSAT-3DR IPWV
over land and oceanic region for all the seasons except Himalayan region and some parts of
Arabian Sea and BoB. This occurred because the infrared and microwave radiometer observations
of land and oceans had been assimilated into the model, which has the higher systematic humidity
when it was compared with Radiosonde data (Andersson et al., 2007). Underestimation of CAMS
IPWV compared with INSAT-3DR over Himalayan region may be due to presence of rugged
terrain/orographic features in the retrieval of IPWV.
RMSE values during winter season ranges (7.5 mm to 13.0 mm) over land region (20º N to 35º N)
and the entire Arabian Sea. Above 35º N latitude including Himalayan region, RMSE values are
less than 7.5 mm. RMSE values ranges (13 mm to 20 mm) observed over the Southern peninsula
of India and BoB region respectively (Figure 8).
RMSE values during pre-monsoon season ranges (2.5 mm to 13.0 mm) over land region (18º N to
40º N), Arabian Sea and Himalayan region observed. RMSE values ranges (13 mm to 20 mm) are over the Southern peninsula of India, Indo Gangetic Plains (IGP) and BoB region respectively (Figure 8).

RMSE values during monsoon season ranges (14. mm to 20.0 mm) over land region (20º N to 35º

N) including North West of Arabian Sea and North East of BoB. Above 35º N latitude, South West

& South East of Arabian Sea including South East of BoB and Himalayan region RMSE values are less than 8.0 mm respectively (Figure 8).

RMSE values during post-monsoon season less than 7.5 mm observed over land region including both Arabian Sea as well as BoB region except Indo Gangetic Plains (IGP) and north East of BoB

ranges (13 mm to 17 mm) respectively (Figure 8).

**RC#** Section 3.4 need to be further improved, particularly about oceanic areas. Also, Figure 9

shows seasonal analyses not annual mean values.

**Response:** Over the oceanic region, seasonal mean IPWV of INSAT-3DR and CAMS  ranges from 25-40 mm (with standard deviation 6-15 mm) and 20-45 mm (SD 6-16 mm) and less than 25

mm with SD of less than 6 mm for both INSAT-3DR and CAMS  IPWV over land region during winter season  respectively (Figure 10).

Over the oceanic region, seasonal mean IPWV of INSAT-3DR and CAMS ranges from 30-45 mm (with standard deviation 7-12 mm) and 35-55 mm (SD 10-16 mm). Over land region, seasonal mean IPWV of INSAT-3DR and CAMS data ranges from 15-38 mm with SD of 2-10 and 20-40

mm with SD of 5-12mm during pre-monsoon season respectively (Figure 10).

Seasonal mean IPWV of INSAT-3DR ranges from 30 mm to more than 60 mm with SD of 2-14

mm and from 50 mm to more than 60 mm with SD of 4-16 mm of CAMS IPWV observed for both land and oceans region during monsoon season respectively (Figure 10).

Over the oceanic region, seasonal mean IPWV of INSAT-3DR and CAMS  ranges from 35-55

mm (with standard deviation 6-10 mm) and 38-55 mm (SD 6-14 mm) and  over land region mean

IPWV of INSAT-3DR and CAMS data ranges from 15-35 mm with SD of 5-12 and 20-40 mm with SD of 10-16 mm during post-monsoon season  respectively (Figure 10).

**RC#** Conclusion section must be improved. Point number four is not demonstrated from the analyses and discussions in the manuscript. Point number five need to be revised because it cannot be understood.

**Response:** Point number four has been removed and Point number five has been modified in the
manuscript.
**RC#** Finally, I recommend that a native English speaker revise the manuscript.
**Response:** Manuscript has been revised as per suggestion by referee.
We once again thank the reviewer for his/her constructive comments/suggestions which made us
to improve the manuscript content significantly.

[revised manuscript text omitted]